# END-TO-END SCHEDULING OF REAL-TIME TASK PIPELINES ON MULTIPROCESSORS

SOHAM SINHA
*Department of Computer Science*
*Boston University, Boston, MA*
*soham1@bu.edu*

RICHARD WEST
*Department of Computer Science*
*Boston University, Boston, MA*
*richwest@bu.edu*

## Abstract

Task pipelines are common in today's embedded systems, as data moves from source to sink in sensing-processing-actuation task chains. A real-time task pipeline is constructed by connecting a series of periodic tasks with data buffers. In a time-critical system, end-to-end timing and data-transfer properties of a task pipeline must be guaranteed. A guarantee could be mathematically expressed by assigning constraints to the tasks of a pipeline. However, deriving task scheduling parameters to meet end-to-end guarantees is an NP-hard constraint optimization problem. Hence, a traditional constraint solver is not a suitable runtime solution.

In this paper, we present a heuristic constraint solver algorithm, *CoPi*, to derive the execution times and periods of pipelined tasks that meet the end-to-end constraints and schedulability requirements. We consider two upper bound constraints on a task pipeline: end-to-end delay and loss-rate. After satisfying these constraints, *CoPi* schedules a pipeline as a set of asynchronous and data independent periodic tasks, under the rate-monotonic scheduling algorithm. Simulations show that *CoPi* has a comparable pipeline acceptance ratio and significantly better runtime than open-source MINLP-solvers. Furthermore, we use *CoPi* to map multiple task pipelines to a multiprocessor system. We demonstrate that a partitioned multiprocessor scheduling algorithm coupled with *CoPi* accommodates dynamically appearing pipelines, while attempting to minimize task migrations.

## 1 Introduction

Real-time, embedded and cyber-physical systems are commonly composed from pipelines of tasks connected by data buffers. In automotive and avionics domains, sensor inputs are fed through a pipeline of processing and control tasks, which ultimately produce actuator outputs. Such time-critical systems benefit from a real-time task pipeline model. Given the increased use of multiprocessors in embedded systems, there is a need to investigate the scheduling of real-time task pipelines on such platforms.

Task pipelines, or cause-effect chains, have received increased attention in recent research work [17,20,32,40,41,60], partly because of their usage in well-known software packages such as ROS [13,65]. Although real-time task pipelines have long been studied [23,49,50], the application of constraints on a pipeline has received little attention [19]. Constraints on a pipeline of periodic tasks ensure that end-to-end properties of a pipeline are guaranteed. However, finding schedulable task runtimes and periods to meet the constraints is an NP-hard problem. Therefore, traditional solvers are unsuitable in runtime scheduling, where tasks and pipelines may dynamically appear in a real-time system.

This paper presents a heuristic constraint solver algorithm for real-time task pipelines, *CoPi*, to derive the runtime budgets and periods of individual pipelined tasks from a list of supplied task budgets. *CoPi* works with two pipeline constraints: the worst-case end-to-end (E2E) delay and loss-rate. The worst-case E2E delay is the maximum time interval between the time a data sample first appears at the input to the first task of a pipeline, and the first time a corresponding output is produced at the last task of the pipeline. The worst-case E2E loss-rate is the number of input messages to the pipeline that do not have a corresponding output with respect to the number of input messages to a pipeline over the period of its first task. As *CoPi* treats pipelined tasks as asynchronous and independent tasks, data might be lost between two communicating tasks if a producer overwrites its output before a consumer has read it. The loss-rate captures how many input messages have no effect at the end of the output of a task pipeline. In addition to these constraints, *CoPi* uses the rate-monotonic scheduling (RMS) algorithm to schedule the tasks, and the RMS utilization upper bound as another constraint. RMS is chosen for its simplicity and popularity in real-time systems.

*CoPi* aims to avoid unnecessary delay and message losses in a pipeline. In previous work [22, 28], tasks are released at offset times, or task precedence relations are created, to mitigate the data dependency between communicating tasks. *CoPi* finds the suitable task runtimes and periods so that no

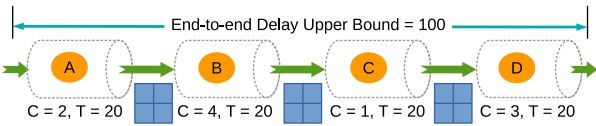

Figure 1: *CoPi* Period Derivation: The above pipeline has an end-to-end delay upper bound constraint of 100 time units. In the simplest case, *CoPi* divides the delay by (number of tasks + 1) [assuming input is available at arbitrary time. If input is only available at the beginning of A's period, then the upper bound could be tightened to 80 time units]. *C* and *T* are respectively the runtime budget and period of a task.

timing and data dependencies occur at runtime between the communicating tasks.

Figure 1 shows a small example where *CoPi* meets the E2E delay upper bound by assigning suitable periods to the pipelined tasks. The example also meets the RMS utilization bound. However, a tighter upper bound on the E2E delay could violate the RMS utilization bound, and *CoPi* needs to find a more appropriate set of task runtimes and periods. *CoPi* tunes the individual task runtimes and periods so that E2E delay and loss-rate are under their upper bounds, while the total utilization does not cross the RMS bound.

As *CoPi* meets the E2E delay and loss-rate guarantees of a pipeline, the asynchronous tasks are scheduled without any timing or data dependencies between each other. We leverage this feature of *CoPi* to map the tasks of *multiple* pipelines to a multiprocessor system. Figure 2 summarizes this main idea. We use the Worst-fit Decreasing (WFD) heuristic to map tasks to processors and also incorporate runtime task migration and scheduling parameter optimization strategies to admit dynamically appearing pipelines.

As real-time systems increasingly operate in dynamic environments (e.g., to perform object detection in autonomous driving [37]), it is likewise increasingly important to handle task pipeline scheduling at runtime. Implementing a complete MINLP solver is difficult and sometimes infeasible in an OS-level scheduler. *CoPi* provides the basis for online practical solutions to pipeline scheduling in real-time multiprocessor systems.

**Summary of contributions:**

1. We formally present the problem of finding suitable runtime budgets and periods of a pipeline of periodic tasks, under two pipeline constraints (E2E delay and loss-rate), and a utilization bound constraint.

2. We propose and analyze a heuristic constraint solver algorithm, *CoPi*, to satisfy all the constraints.

3. We demonstrate the usefulness of *CoPi* by using it in a multiprocessor scheduling algorithm, which includes runtime task migration and pipeline scheduling optimization mechanisms to map dynamic pipelines to processor cores.

4. We evaluate *CoPi* against open-source Mixed-integer Nonlinear Programming (MINLP) solvers such as GEKKO [5], scipy [2] and pyomo [11,34] with simulated task pipelines[1]. We show that *CoPi* performs up to an order of magnitude better in runtime, and comparably in pipeline acceptance ratio (i.e., ratio of the number of schedulable pipelines by a solver to the total number of pipelines), with respect to other solvers. We have also tested *CoPi* with tasksets

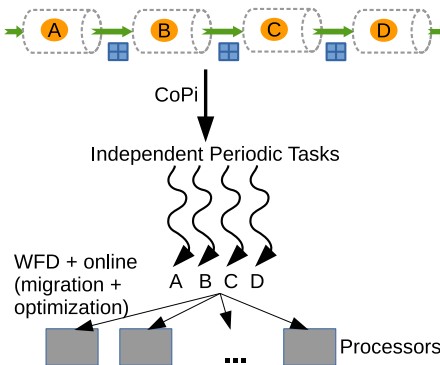

Figure 2: *CoPi* converts a task pipeline to a set of independent and asynchronous tasks. A four-slot asynchronous buffer [62] is used between a pair of communicating tasks. The tasks are then mapped to a multiprocessor system.

from the WATERS 2015 workshop paper [43] and observe similar performance to when pipelines are randomly generated. Moreover, simulation experiments for multiprocessor scheduling demonstrate *CoPi*'s usefulness in maximizing processor utilization and minimizing runtime task migrations.

The next section describes the system model. Then, Section 3 and 4 formally define the pipeline constraints and the problem, respectively. Section 5 describes and analyzes *CoPi*. Section 6 explains the multiprocessor scheduling algorithm using *CoPi* and two pipeline acceptance improvement techniques. An evaluation with simulated task pipelines is presented in Section 7. Then, we explore a few possible future implementation opportunities of *CoPi* in RTOS schedulers in Section 8. Related Work is discussed in Section 9, followed by conclusions and future work in Section 10.

## 2   System Model

In this section, we define the task, pipeline and scheduling models in the system.

### 2.1   Task Model

All tasks are assumed to be asynchronous, which means they do not wait or block on other tasks for a resource. Every task has a four-slot asynchronous buffer [62] for its input, and another for its output, and works with the most recent available data. A task $\tau$ in the system is defined by a two-tuple $(C, T)$, as follows:

- $C$: the worst-case runtime budget or capacity[2] of a task to read a message (or *data-unit*) from its input buffer, process the data and write a message (or data-unit) to its output

---

[1]The artifacts are available at https://github.com/sohamm17/pipe_schedule

[2]We use the terms *runtime budget*, *execution time budget*, *budget* and *capacity* interchangeably.

buffer. This is extensible to inputs or outputs of more than a single data-unit in a four-slot buffer.

- $T$: the period and deadline of an implicit-deadline periodic task. In every new period, $\tau$ works on new data, and generates a unique output.

$C$ is the *initial* runtime budget to process a single data-unit by a pipelined task. Later, we show how our heuristic constraint solver algorithm *CoPi* adjusts the final allocated budget using $C$ to meet the end-to-end constraints.

## 2.2   Pipeline Model

A pipeline $P$ is represented by an ordered set of periodic tasks: $S = \{\tau_1, ..., \tau_N\}$. The cardinality of $S$ is $N$. $\forall \tau_i, \tau_j \in S$, $i < j$ implies that data flows from $\tau_i$ to $\tau_j$. Without the loss of generality, we consider unidirectional pipelines without cycles.

### 2.2.1   Overview of Pipeline Constraints

We explain the pipeline constraints formally in Section 3. We provide a high-level overview of them below:

- $E$: the worst-case end-to-end latency or delay of a pipeline i.e., the maximum time a single message takes from the input to the output of a pipeline. The input appears at any arbitrary time for the first task $\tau_1$ of a pipeline.

- $L$: the end-to-end loss-rate i.e., the number of input messages that do not have a corresponding output message, for every input message to the pipeline, over the period of its first task. It is expressed as a fraction or a percentage. As the tasks work with asynchronous buffers, a certain message might be overwritten and lost due to more than one consecutive write by a producer task before a read by a consumer task. $L$ captures how many messages are lost per input message.

### 2.2.2   Communication Model

A task communicates with another task with a message or data unit. In practice, a message is either a sensor input such as IMU data, an actuator output like steering control, or processed data in between inputs and outputs. In the automotive and factory automation industries, messages are also called labels [43].

The tasks in a pipeline communicate with each other following an implicit communication model [33]. Therefore, a message is read from a shared input buffer at the beginning of a job of a task, and used throughout the task before writing to a shared output buffer. This ensures that a single and consistent copy of a message is used for a single job invocation of a task.

Simpson's four-slot algorithm [62] is used to exchange data between a pair of communicating tasks via a register-based fully asynchronous buffer. In this algorithm, two pairs of slots are maintained separately for a reader and a writer. The writer uses two control bits to indicate which pair and slot are being most recently written. The reader uses another control bit to indicate which pair it is reading. The algorithm shows that four slots are enough for a reader and a writer to communicate asynchronously with each other [58, 62] while ensuring messages are exchanged as integral units.

$\tau_1$, the first task of a pipeline, is the source task. $\tau_1 = \tau_{src}$. $\tau_N$, the last task of a pipeline, is the sink task. $\tau_N = \tau_{sink}$. $\tau_1$ does not wait or block for its input data because we assume that an input is always available for $\tau_1$. It is realistic since the source generally reads from a sensor input or digital media. In absence of new input data, the source task sends the most recently available data [16]. The same assumption applies for the output of a sink task.

## 2.3   Scheduling Model

The system schedules all the tasks using the rate-monotonic scheduling (RMS) algorithm [48]. We assume that each periodic task $\tau_i$, running in a sporadic server [64] with a processor capacity reserve [51], will have a maximum runtime of $C_i$ time-units in every $T_i$ time-units, as it is designed to be implemented in an RTOS like Quest [18]. We choose RMS because it is a low-overhead, fixed-priority scheduling algorithm that is popular in many RTOSs [4, 10, 18].

Each task is assigned a priority by the RMS algorithm, which is inversely proportional to its period. If two tasks of the same pipeline have the same periods, then the preceding task in the pipeline is given higher priority. In other words, $prio(\tau_i) > prio(\tau_j)$, if $i < j$ and $T_i = T_j$. $prio(\tau_i)$ is the priority of task assigned by RMS.

If a task has already finished its work for a job invocation, it yields and does not start its next job until the next period. This ensures that a single job invocation of a task does not overwrite its already written output in an asynchronous communication. Moreover, the fixed execution time tightly bounds a pipeline's end-to-end latency [32].

## 3   Pipeline Constraints

We consider constraints on the two pipeline parameters and on the total task utilization. The two pipeline parameters, end-to-end delay and loss-rate, are computed from the ordered taskset $S$. We first discuss a computational analysis of the parameters, and then present the constraints on them.

## 3.1   End-to-end Delay ($E$) Computation

The worst-case end-to-end delay of a pipeline is the maximum time for a message to appear at $\tau_{src}$ and emit from $\tau_{sink}$. It

is also known as the maximum reaction time [16, 22] in a cause-effect chain [3, 7].

Davare *et al.* presented the first but conservative upper bound on the worst-case end-to-end delay for a pipeline of periodic tasks with arbitrary budgets and periods [19]. If $R_i$ is the worst-case response-time of $\tau_i$, then the worst-case end-to-end delay is the following:

$$E = \sum_{i=1}^{N}(T_i + R_i) \tag{1}$$

In the above equation, $R_i$ is recursively calculated by initially estimating to be equal to the task period $T_i$ for each task $\tau_i$ [39]. As Equation 1 is a recursive equation, the time-complexity of computing the equation depends on the wanted precision on $E$. Response-time calculation for fixed-priority scheduling is known to be NP-hard [21]. Nevertheless, a bounded computation time is preferred in a runtime task scheduling algorithm. Therefore, $R_i$ is replaced in the above equation with $T_i$. If $\tau_i$ is feasibly scheduled, then $R_i$ is less than or equal to $T_i$. Therefore, a faster computable version of Equation 1 is the following:

$$E = 2 \times \sum_{i=1}^{N} T_i \tag{2}$$

In offline or slower design-time analysis, Equation 1 is tolerable. For use in runtime scheduling, Equation 2 is preferable.

Dürr *et al.* tightened Equation 1 by considering task priorities between pairs of communicating tasks in a pipeline [20]. They use forward and backward cause-effect chains to derive a tighter upper bound on E2E delay. After converting the sporadic task model to the periodic task model as done in a subsequent work [32], the worst-case E2E delay (also called the *maximum reaction time*) of a pipeline considering all the tasks are released at the critical instant [48] (initial release offset is 0), as proved by Dürr *et al.*, is the following:

$$E \leq T_1 + R_N + \sum_{i=1}^{N-1} max(R_i, T_{i+1} + R_i \times I) \tag{3}$$

In above equation, $I$ is the Iverson bracket. $I = 1$, if the $(i+1)^{th}$ task has higher priority than $i^{th}$ task. $I = 0$, otherwise.

As it is done for Equation 1 and 2, the following equation is a conservative but faster computable version of Equation 3:

$$E \leq T_1 + T_N + \sum_{i=1}^{N-1} max(T_i, T_{i+1} + T_i \times I) \tag{4}$$

The asymptotic time-complexity of Equations 2 and 4 is $O(N)$. Equation 2 and 4 are useful in designing a runtime end-to-end pipeline scheduling algorithm. In this paper, Equation 4 is used for the uniprocessor pipeline scheduling. For multiprocessor scheduling, Davare *et al.*'s Equation 2 is used to avoid dependencies on task priorities in a pipeline, as a pipelined task could be mapped to any processor.

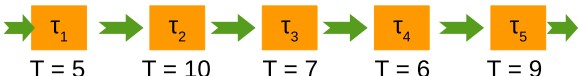

Figure 3: End-to-end delay example.

### 3.1.1 Example

Figure 3 shows an example pipeline. The end-to-end delays are 74 and 63, respectively with Equation 2 and Equation 4, if the tasks are scheduled with RMS. The calculation of Equation 4 for the example is the following:

$$\begin{aligned}
E &\leq 5 + 9 + (max(5, 10 + 5 \times 0) + max(10, 7 + 10 \times 1) + \\
&\quad max(7, 6 + 7 \times 1) + max(6, 9 + 6 \times 0)) \\
&\leq 5 + 9 + (10 + (10 + 7) + (7 + 6) + 9) \\
&\leq 63
\end{aligned}$$

## 3.2 End-to-end Loss-rate ($L$) Computation

Data loss is an issue in systems involving sensors and actuators [30, 42, 72] The end-to-end loss-rate of a pipeline is the number of input messages that do not have a corresponding output, per input message to a pipeline, over the period of its first task. Suppose the total number of input messages per period of the first task of a pipeline is $I$, and the number of corresponding output messages for $I$ inputs is $O$, then loss-rate is defined by Equation 5. If $O$ is greater than or equal to $I$, then no messages are lost, and the loss-rate is deemed 0. Loss-rate can also be realized in terms of Feiertag *et al.*'s concept of reachability [22], where it is the ratio of *non-reachable* messages to the total number of input messages per period of the first task of a pipeline.

$$\begin{aligned}
L &= \frac{I - O}{I}, \text{if } O < I \\
&= 0, \text{if } O \geq I
\end{aligned} \tag{5}$$

Input and output messages are usually generated and sent from a sensor to an actuator, associated with an I/O buffer. However, an input may not come from an external buffer or input device, but may simply be generated by the source task of a pipeline. In that case, the generated messages are considered to be the inputs to a pipeline.

To calculate the loss-rate, we assume that the pipelined tasks are feasibly scheduled using a real-time scheduling algorithm following the scheduling model described in Section 2.3. For every pair of producer-consumer tasks ($\tau_p \rightarrow \tau_c$) in a pipeline, the consumer could either oversample ($T_c \leq T_p$) or undersample ($T_c > T_p$) its input from its corresponding producer. Based on the relationships between the periods of all producer-consumer pairs starting from the source task to the sink task, the end-to-end loss-rate of a pipeline is calculated. The calculation is explained later in this section.

### 3.2.1    Sampling Ratio

We define the sampling ratio $f_{\tau_p \to \tau_c}$ of a producer-consumer pair ($\tau_p \to \tau_c$) as the number of output messages of $\tau_c$ per unique input message of $\tau_p$. According to the task and scheduling model, a task generates a single message during its runtime budget per period, and retires until its next invocation. Therefore, $f_{\tau_p \to \tau_c}$ is calculated from the producer's period divided by the consumer's period:

$$f_{\tau_p \to \tau_c} = \frac{T_p}{T_c} \qquad (6)$$

Then, the loss-rate of a producer-consumer pair is $(1 - f_{\tau_p \to \tau_c}) = (1 - \frac{T_p}{T_c})$, if $f_{\tau_p \to \tau_c} < 1$, 0 otherwise. Examples are given later in the section.

**Oversampling**    In the case of an oversampling consumer ($T_p \geq T_c$), the data from the producer will be overrepresented in the output by the consumer. For example, consider $\tau_p = (C_p = 2, T_p = 40), \tau_c = (C_c = 1, T_c = 10)$. $\tau_p$ runs for 2 time-units in every 40 time-units and reads, processes, and writes a single input message. $\tau_c$ does the same in 1 time-unit in every 10 time-units. Therefore, $\tau_c$ will emit the same output 4 times for a unique input of $\tau_p$. Hence, the sampling ratio is $\frac{T_p}{T_c} = 4$. Therefore, the *oversampling ratio* is: $O_{\tau_p \to \tau_c} = f_{\tau_p \to \tau_c} = \frac{T_p}{T_c} \geq 1$. The loss-rate is 0 in this case as the sampling ratio is more than 1. This means that no messages are lost in this producer-consumer pair.

**Undersampling**    The case of an undersampling consumer ($T_p < T_c$) is more nuanced because data might be lost. The data from a producer might be overwritten before a consumer has read it, as the consumer has a larger period. For example, consider $\tau_p = (C_p = 1, T_p = 10), \tau_c = (C_c = 5, T_c = 40)$. $\tau_p$ takes 1 time-unit in every 10 time-units to read, process and finally output a message for $\tau_c$. $\tau_c$ takes 5 time-units in every 40 time-units to read a single message from $\tau_p$, process it, and write its own single output message. Therefore, $\tau_p$ will run 4 times and produce 4 unique messages in 40 time-units. However, $\tau_c$ only runs once in 40 time-units and works with only 1 of 4 messages produced by $\tau_p$. Therefore, the sampling ratio is $\frac{T_p}{T_c} = \frac{1}{4}$. Here, the *undersampling ratio*: $U_{\tau_p \to \tau_c} = f_{\tau_p \to \tau_c} = \frac{T_p}{T_c} < 1$, yielding a loss-rate of $(1 - \frac{1}{4}) = \frac{3}{4}$ or 75%.

**Pipeline Sampling Ratio**    Let's consider a pipeline $P_x = \{S_x\}$. Subscript $x$ is used to distinguish a pipeline. $P_x$ has two communicating tasks ($S_x = \{\tau_1, \tau_2\}, N_x = 2$) i.e., a single producer-consumer pair ($\tau_1 \to \tau_2$). The *sampling ratio* is $f_x = \frac{T_1}{T_2}$. The whole pipeline is *oversampled* if $f_x \geq 1$ and *undersampled* if $f_x < 1$.

Now, consider that $P_x$ is extended by adding a new task $\tau_{new}$ at the end of $P_x$. A new pipeline $P_y$ is thus formed whose ordered taskset $S_y$ is $\{\tau_1, \tau_2, \tau_{new}\}$, and length is $N_y = N_x + 1 = 3$.

Table 1: Rules to calculate the lower bound on the sampling ratio of a resultant pipeline $P_y$ after adding a new $\tau_{new}$ task to a pipeline $P_x$.

| Rule | $P_x(\tau_1 \rightarrow \dots \rightarrow \tau_{N_x})$ | $\tau_{new}$ | Lower Bound on new Sampling Ratio ($f_y$) |
|------|------|------|------|
| 1 | Undersampled | Oversampled | $f_x$ |
| 2 | Undersampled | Undersampled | |
| 3 | Oversampled | Oversampled | $f_x \times \dfrac{T_{N_x}}{T_{new}}$ |
| 4 | Oversampled | Undersampled | |

$\tau_{new}$ could be oversampling or undersampling compared to the last task in $P_x$, $\tau_2$ or $\tau_{N_x}$.

We want to calculate a lower bound on the sampling ratio to derive an upper bound on the loss-rate. We define 4 rules to calculate a lower bound on the sampling ratio of a pipeline. The rules are summarized in Table 1 and proved below:

**Rule 1**    *If a pipeline $P_x$ is undersampled, adding an oversampling task whose period is smaller than the period of $P_x$'s last task $\tau_{N_x}$, does not change the lower bound of the resultant pipeline's sampling ratio.*

*Proof.* Let's take an undersampling pipeline $P_x$. Its sampling ratio is $f_x = \frac{B}{A}$. It produces $B$ output messages for every $A$ input messages to a pipeline. $B < A$.

A new oversampling $\tau_{new}$ task is added to $P_x$ to form pipeline $P_y$. Therefore, $\tau_{new}$'s period is less than or equal to the period of $P_x$'s last task $\tau_{N_x}$. $T_{new} \leq T_{N_x}$.

For every new input message to $\tau_{new}$, $O_{\tau_{N_x} \to \tau_{new}} = (\frac{T_{N_x}}{T_{new}} \geq 1)$ number of output messages are produced by $\tau_{new}$. Therefore, for every $B$ outputs from $P_x$ to $\tau_{new}$, $(B \times O_{\tau_{N_x} \to \tau_{new}})$ outputs are produced by $\tau_{new}$.

However, $(B - A)$ number of messages are already lost in the pipeline $P_x$. The oversampling task $\tau_{new}$ cannot recover those messages. Therefore, $(B \times O_{\tau_{N_x} \to \tau_{new}})$ messages just represent the oversampled messages produced by task $\tau_{new}$. The number of unique output messages per input message of the pipeline remains same. Therefore, $P_x$'s sampling ratio ($f_x$) remains the lower bound of $P_y$'s sampling ratio.    ∎

**Example**    Consider the example given in Figure 4. $P_x = \{S_x = (\tau_1, \tau_2)\}$. $N_x = 2$. $P_y$ is formed by adding $\tau_3$ to $P_x$. $\tau_1$'s input is given at the left most side. Each line represents a unique message with a corresponding ID shown by the labels 1 to 4. $\tau_1$ reads one message in its single job invocation, increases the ASCII value of the input and writes to its output buffer. $\tau_2$ and $\tau_3$ simply read one input message at a time and forward it through to the corresponding output buffer.

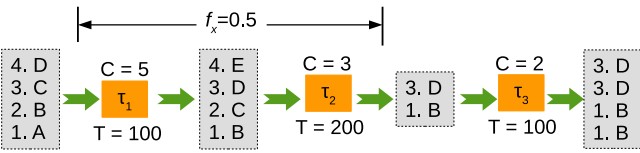

Figure 4: Example of Rule 1 of the pipeline sampling ratio calculation.

$P_x$'s sampling ratio $f_x$ is $\frac{T_1}{T_2} = 0.5$. We see that $\tau_2$ emits its $1^{st}$ (B) and $3^{rd}$ (D) input messages, although it receives B, C, D, E as inputs. $\tau_2$ emits one out of every two input messages. Now, we add $\tau_3$ after $\tau_2$. $\tau_3$ is oversampling with respect to $\tau_2$, exemplifying *Rule 1*. As $\tau_3$ runs twice as frequently as $\tau_2$, it replicates one input message two times in its output. Therefore, it emits B, B, D, D for B, D inputs coming from $\tau_2$. However, the repetitions do not recover the lost messages A, C. Therefore, the lower bound of the sampling ratio of the new pipeline $P_y$ remains the sampling ratio of pipeline $P_x$. In this case, that is 0.5.

**Rule 2**  *If an undersampling task $\tau_{new}$ is added at the end of an undersampled pipeline $P_x$, then the resultant sampling ratio is lower bounded by the undersampling ratio of $P_x$, $f_x$, multiplied by the undersampling ratio of the last task of $P_x$ and the new task .*

*Proof.* Let's take the same undersampling pipeline $P_x$ from *Rule 1*. $f_x = \frac{B}{A}$ and $B < A$. An undersampling $\tau_{new}$ task is added to $P_x$ to form pipeline $P_y$. Therefore, $\tau_{new}$'s period is greater than the period of $P_x$'s last task $\tau_{N_x}$. $T_{new} > T_{N_x}$.

For every new input message to $\tau_{new}$, $\frac{T_{N_x}}{T_{new}} = U_{\tau_{N_x} \to \tau_{new}} (< 1)$ outputs are produced by $\tau_{new}$. Therefore, for every $B$ inputs to $\tau_{new}$, only $B \times U_{\tau_{N_x} \to \tau_{new}}$ messages are produced. Hence, the sampling ratio of the newly formed pipeline $P_y$: $f_y = \frac{B \times U_{\tau_{N_x} \to \tau_{new}}}{A} = f_x \times \frac{T_{N_x}}{T_{new}}$. $f_y$ provides the lower bound for the new pipeline $P_y$. ∎

**Example**  Consider the same example given in Figure 4 but with period of $\tau_3 = 400$. In this case. $\tau_3$ emits only the $1^{st}$ (B) message. Therefore, the sampling ratio is $0.5 \times \frac{200}{400} = 0.25$.

**Rule 3**  *If an oversampling task $\tau_{new}$ is added at the end of an oversampled pipeline $P_x$, then the resultant sampling ratio is lower bounded by $f_x$ multiplied by the oversampling ratio of the last task of $P_x$ and the new task.*

*Proof.* Let's consider an oversampling pipeline $P_x$. $f_x = \frac{B}{A}$ and $B \geq A$. An oversampling task $\tau_{new}$ is added to $P_x$ to form pipeline $P_y$. $T_{new} \leq T_{N_x}$. For every $B$ outputs from $P_x$ to $\tau_{new}$, $(B \times O_{\tau_{N_x} \to \tau_{new}})$ outputs are produced by $\tau_{new}$.

Therefore, the sampling ratio of the newly formed pipeline $P_y$: $f_y = \frac{B \times O_{\tau_{N_x} \to \tau_{new}}}{A} = f_x \times \frac{T_{N_x}}{T_{new}}$. $f_y$ is the lower bound on the sampling ratio of $P_y$. ∎

**Example**  Consider the example in Figure 5, which is similar to Figure 4 but with different periods. The sampling ratio after adding $\tau_3$ at the end of $P_x (S_x = \{\tau_1, \tau_2\})$ is 4. Here, 4 accurately represents the oversampling ratio.

**Rule 4**  *If an undersampling task $\tau_{new}$ is added to an oversampled pipeline $P_x$, the resultant sampling ratio is $f_x$ multi-*

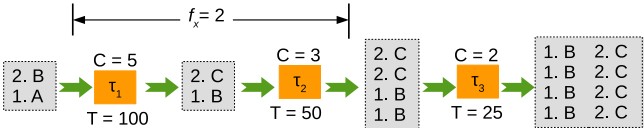

Figure 5: Example of Rule 3 of the pipeline sampling ratio calculation.

*plied by the undersampling ratio of $P_x$'s last task and the new task .*

*Proof.* Let's consider the same oversampling pipeline from the above rule, $P_x$. $f_x = \frac{B}{A} \geq 1$. An undersampling task $\tau_{new}$ is added to $P_x$ to form pipeline $P_y$. $T_{new} > T_{N_x}$. For every $B$ input messages to $\tau_{new}$, it produces only $(B \times U_{\tau_{N_x} \to \tau_{new}})$ messages, which is less than $B$, as $U_{\tau_{N_x} \to \tau_{new}} < 1$. We already know that $P_x$ produces $f_x$ output messages for every single input to it.

Therefore, the sampling ratio of the newly formed pipeline $P_y$: $f_y = f_x \times U_{\tau_{N_x} \to \tau_{new}} = f_x \times \frac{T_{N_x}}{T_{new}}$. $f_y$ also provides the lower bound on the sampling ratio of $P_y$. ∎

**Example**  Consider the example in Figure 5 but with $T_3 = 200$. The sampling ratio between $\tau_2$ and $\tau_3$ is $\frac{50}{200} = \frac{1}{4}$. Therefore, $\tau_3$ is able to output only 1 message for every 4 input messages coming from $\tau_2$. So, it will only output B in the example. Therefore, the lower bound on the full pipeline's sampling ratio is $f_x \times \frac{1}{4} = 2 \times \frac{1}{4} = \frac{1}{2}$. As we can see that two messages (A, B) were input to the pipeline, but only 1 message was output.

Following the above rules, the sampling ratio of a pipeline could be recursively derived by calculating the sampling ratio from the source task of a pipeline until the sink, treating the next task in a pipeline as $\tau_{new}$. The first producer-consumer pair is considered a pipeline $P_x$. Then, a new task is added at the end of the $P_x$ pipeline, and a new sampling ratio is calculated. The end-to-end sampling ratio is computed by the end of the overall pipeline, when there are no more tasks.

### 3.2.2  Upper Bound on Loss-rate from Sampling Ratio

The upper bound on a pipeline $P$'s loss-rate is expressed in terms of its sampling ratio $f$ as follows:

$$L = 0, \text{if } f_P \geq 1$$
$$\leq 1 - f_P, \text{if } f_P < 1 \tag{7}$$

## 3.3  Formalization of Pipeline Constraints

Equations 4 and 7 show the upper bounds of the end-to-end delay and loss-rate of a pipeline. A task's runtime budget $C$ to read, process and write 1 message is usually determined by performing a worst-case execution time (WCET) analysis [68]. A pipeline is then constructed by chaining up these periodic tasks. End-to-end delay and loss-rate constraints are applied on a pipeline to guarantee a certain level of quality of service. The unknown variables here are the periods of the

individual tasks of a pipeline, and, thus, the problem is to find them. The constraints are summarized below:

1. $E$ should be upper bounded by a constant $E^{UB}$, meaning that the worst-case end-to-end delay computed from Equation 4 should not exceed $E^{UB}$.

2. $L$ should be upper bounded by $L^{UB}$, meaning that the loss-rate computed from Equation 7 should not exceed $L^{UB}$.

3. The total pipeline utilization should be within a constant upper bound of $U^{UB}$. This constraint comes from the scheduling model, in our case, the RMS bound (See Section 2.3).

*Constraints 1, 2* and *3* for a pipeline are respectively formalized in Equations 8, 9, and 10

$$T_1 + T_N + \sum_{i=1}^{N-1} max(T_i, T_{i+1} + T_i \times I) \leq E^{UB} \qquad (8)$$

$$L \leq L^{UB} \qquad (9)$$

$$\sum_{\forall \tau_i \in S} \frac{C_i}{T_i} \leq U^{UB} \qquad (10)$$

## 4 Problem Statement

Given a list of budgets for a set of pipelined tasks ($C_i, \forall \tau_i \in S$) and a set of constant upper bounds ($E^{UB}, L^{UB}, U^{UB}$) as constraints, the challenge is to find suitable periods, $T_1, T_2, \ldots, T_N$, such that the constraints are satisfied.

To be precise, Equations 8, 9 and 10 need to be satisfied to find a set of suitable periods. This is a constraint programming problem that is in the class of integer nonlinear programming problems, assuming integer task periods. It is a nonlinear programming problem because of Equation 10 where the period is in the denominator. The problem is known to be NP-hard [36].

### 4.1 Budget Adjustment

Until now, we have considered the task runtime budget to be a constant $C$ in the optimization problem. This requirement could be relaxed by allowing increments of task budgets in integer multiples of $C$. If a task's budget is decreased from the initial budget $C$, it would be unable to read, process and write 1 message or data-unit, assuming that a fragment of 1 data-unit is invalid.

When a task $\tau$'s budget is increased from $C$ to $MC$ ($M$ is an integer constant greater than 1), it means $\tau$ processes $M$ messages in its single job invocation. $\tau$'s input and output buffer sizes are also increased to pass $M$ messages in each slot of the four-slot buffer.

### 4.1.1 Additional Constraint

The constraints for the budget adjustment are the following:

$$M_i \geq 1 \qquad (11)$$

$$\forall_{\tau_i \in S} \ FC_i = M_i \times C_i \qquad (12)$$

$FC_i$ is the final allocated runtime budget of $\tau_i$. $M_i$ is called a budget multiplier.

Our heuristic constraint solver algorithm, *CoPi*, uses the budget adjustment mechanism to bound the other pipeline constraints. The details are in Section 5.2.2. However, these budget adjustment constraint equations are not supplied to the open-source solvers, by default. Therefore, they use only Equations 8, 9 and 10, unless otherwise specified.

### 4.1.2 Loss-rate Recalculation after Budget Adjustment

The sampling ratio formula is extended from Equation 6 to accommodate the budget adjustment by *CoPi*:

$$f_{\tau_p \to \tau_c} = \frac{T_p}{T_c} \times \frac{M_c}{M_p} \qquad (13)$$

$M_c$ and $M_p$ are respectively the consumer and producer budget multipliers from Equation 11. New loss-rate is calculated from the adjusted sampling ratio.

### 4.2 MINLP Solvers

We have modeled the pipeline constraints in three open-source Mixed-Integer Non-Linear Programming (MINLP) solvers, written in Python: GEKKO [5], pyomo [11, 34] and scipy [2]. We compare their performances to our heuristic constraint solver, *CoPi*, in terms of the number of accepted task pipelines in Section 7. We have also considered GNU Linear Programming Kit [29], Google-OR Tools [31] and other solvers, but they lack integer nonlinear programming features, and the above ones suffice for the purpose of this work.

## 5 Pipeline Constraint Solver Heuristic

In this section, we explain the details of our heuristic constraint solver algorithm for uniprocessor scheduling.

### 5.1 *CoPi*'s Objective and Approach

The primary objective of *CoPi*, our constraint solver heuristic for end-to-end scheduling of a real-time task pipeline, is to avoid unnecessary delay and data loss among the communicating tasks. Once the data-dependencies between the pipelined tasks are handled by tuning the task parameters, all the tasks run independent of each other without waiting for job release and completion times [7, 17].

Given the initial task budgets, the upper bounds of the pipeline parameters (E2E delay and loss-rate) and the RMS utilization bound, *CoPi* derives the task periods and new runtime budgets. Gerber *et al.* also proposed a similar approach of deriving task periods, offsets and other parameters from the end-to-end constraints, albeit on task precedence relations [28]. We utilize the core idea of deriving suitable budgets and periods from the end-to-end requirements, so that the pipelined tasks could be independently executed.

For a multiprocessor system, runtime task migrations are feasible because of *CoPi*'s conversion of pipelined tasks to independent asynchronous tasks (see Figure 2). Although process-to-core mapping and migration should also consider cache, memory and other microarchitectural properties, these should be handled at the system implementation level, and included in an extended task and scheduling model.

## 5.2 *CoPi* Heuristic Algorithm

Pseudocode for *CoPi* is provided in Algorithm 1. It finds the suitable periods of a pipeline of $N$ tasks under all the constraints (*Constraint 1, 2* and *3* from Section 3.3).

*CoPi* takes the initial task runtime budgets ($C$ in the model and `budgets` in Algorithm 1) and the desired upper bound on the end-to-end delay and loss-rate ($E^{UB}$ and $L^{UB}$ in the model, and `e2e_ub` and `lr_ub` in Algorithm 1) as its inputs. `budgets` are given in the same order as in the ordered taskset $S$ in the pipeline model. $\alpha$ and $\beta$ are *CoPi*'s internal tuning parameters that are also taken as inputs and explained later.

### 5.2.1 Stage 1

Lines 8–13 in Algorithm 1 show Stage 1. *CoPi* starts by setting all the task periods to be the same in Line 9: $eq\_period = \frac{e2e\_ub}{N+1}$, where $N$ is the pipeline length. By trying to assign the same equal period to all the tasks, *CoPi* tries to eliminate any loss between the pipelined tasks. Thus, any loss-rate upper bound constraint is satisfied, as the loss-rate is 0 for equal task periods.

To satisfy the end-to-end delay constraint, `e2e_ub` is divided by $(N+1)$ instead of $N$. As per the scheduling model in Section 2.3, the earlier appearing task in a pipeline is given higher priority in RMS algorithm for tasks with equal periods. Therefore, the upper bound on end-to-end delay following Dürr *et al.*'s Equation 4 is: $E \leq \left( (N+1) \times eq\_period \right)$ [20]. Thus, the end-to-end delay constraint is implicitly satisfied by the choice of equal task periods of $\left( \frac{e2e\_ub}{N+1} \right)$: $E \leq \left( (N+1) \times eq\_period \right) = e2e\_ub$.

As both the pipeline constraints are satisfied, *CoPi* checks the schedulability of the task pipeline with the RMS utilization bound constraint in Line 11. The call to `utilization_bound_test(taskset)` using RMS performs the following check: $\left( \sum_{i=1}^{N} \frac{C_i}{T_i} \leq N \times (2^{\frac{1}{N}} - 1) \right)$.

---

**Algorithm 1** *CoPi* Pipeline Constraint Solver

1: **Input:** *budgets*[$N$] - Budgets of $N$ Pipelined Tasks in an ordered sequence from source to destination
2: **Input:** *e2e_ub* - upper bound of end-to-end delay
3: **Input:** *lr_ub* - upper bound of end-to-end loss-rate
4: **Input:** *util_ub* - upper bound on processor utilization. Used if less than the RMS utilization bound.
5: **Input:** $\alpha$ - The multiplicative scaling factor
6: **Input:** $\beta$ - The divisive scaling factor
7: **Output:** If schedulable: an ordered taskset with budgets and periods, else: not schedulable.
8: `// Stage 1`
9: $eq\_period = \frac{e2e\_ub}{N+1}$
10: $taskset = [(b, eq\_period)$ for $b$ in $budgets]$
11: **if** `utilization_bound_test(taskset)` **then**
12:     **return** *taskset*
13: **end if**
14: `// Stage 2`
15: $scaled\_period = \alpha \times eq\_period$
16: $taskset = [(b, scaled\_period)$ for $b$ in $budgets]$
17: **while** *True* **do**
18:     $one\_pipe\_changed = False$
19:     **for** $i = 0$ to $N - 2$ **do**
20:         $producer = taskset[i]$
21:         $consumer = taskset[i+1]$
22:         **if** $producer.budget < \frac{producer.period}{\beta}$ and $consumer.budget \times \beta < consumer.period$ **then**
23:             $producer = (producer.budget, \frac{producer.period}{\beta})$
24:             $consumer = (\beta \times consumer.budget, consumer.period)$
25:             **if** `utilization_bound_test (taskset)` **then**
26:                 $one\_pipe\_changed = True$
27:                 **if** `total_e2e_delay(taskset)` $\leq$ *e2e_ub* and `loss_rate(taskset)` $\leq lr\_ub$ **then**
28:                     **return** *taskset*
29:                 **end if**
30:             **end if**
31:         **end if**
32:     **end for**
33:     **if** not $one\_pipe\_changed$ **then**
34:         break
35:     **end if**
36: **end while**
37: `// Stage 3`
38: **for** $i = N - 1$ to $0$ **do**
39:     $cur\_task = taskset[i]$
40:     $cur\_budget = cur\_task.budget$
41:     $cur\_period = cur\_task.period$
42:     $init\_budget = cur\_task.init\_budget$
43:     **while** $\frac{cur\_budget}{\beta} \geq init\_budget$ **do**
44:         $cur\_budget = \frac{cur\_budget}{\beta}$
45:         $cur\_period = \frac{cur\_period}{\beta}$
46:     **end while**
47:     $taskset[i] = (cur\_budget, cur\_period)$
48:     **if** `utilization_bound_test(taskset)` and `total_e2e_delay(taskset)` $\leq e2e\_ub$ and `loss_rate(taskset)` $\leq lr\_ub$ **then**
49:         **return** *taskset*
50:     **end if**
51: **end for**
52: **return** none

---

For a large value of *e2e_ub*, Stage 1 may return a schedulable pipeline, as the total utilization of the pipelined tasks is relatively small. For smaller and tighter values of *e2e_ub*, *CoPi* moves on to the Stage 2.

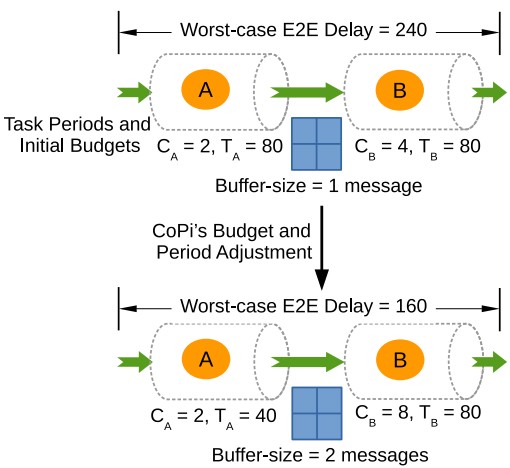

Figure 6: Period and budget adjustment example.

### 5.2.2  Stage 2

Lines 14–36 in Algorithm 1 show Stage 2. In this stage, *CoPi* tries adjusting the task periods to bring down the total utilization while satisfying the end-to-end delay and loss-rate constraints. In order to do so, in step 1, it scales up all the task periods by a constant factor ($\alpha$) to reduce the total task utilization. However, that increases the end-to-end delay and violates the constraint. Then in step 2, *CoPi* considers all the producer-consumer pairs one by one. It scales down the period of a producer by a constant integer factor ($\beta$) and scales up the runtime budget of a corresponding consumer by the same factor $\beta$, in an effort to bring down the end-to-end delay and keep the loss-rate under constraint.

**Rate-matching Heuristic**  Before explaining Stage 2 in Algorithm 1, we explain how *CoPi* adjusts the budget and periods. To reiterate from Section 4.1, when *CoPi* changes a task $\tau$'s runtime budget from $C$ to $(\beta \times C)$, it implies that the task now reads, processes and writes $\beta$ number of messages or data-units in a single job invocation.

Consider an example given in Figure 6. A pipeline is shown with $S = \{A, B\}$ and their initial runtime budgets of 2 and 4, respectively. Imagine after step 1 of stage 2, *CoPi* assigns period of 80 to both the tasks. This pipeline is shown at the top of Figure 6. The end-to-end delay of this pipeline is 240, as calculated using Equation 4.

Then, *CoPi* divides the producer $A$'s period by $\beta$ and also multiplies the consumer $B$'s budget by $\beta$, with $\beta$=2. The resultant pipeline is at the bottom of Figure 6. The end-to-end delay is reduced to 160 from 240 time units. In the new pipeline, $B$ will consume 2 messages in its single job invocation, with a runtime budget of 8. $A$ will still produce 1 message in its runtime because its budget is unchanged. However, $A$'s period is halved, so it will run twice within a single period of $B$. Therefore, the capacity of the four-slot asynchronous buffer is increased to store 2 messages from $A$. In its single job invocation, $B$ will now consume those 2

messages. Therefore, the final sampling rate of the pipeline is still $\left(\frac{40}{80} \times \frac{2}{1}\right) = 1$.

By adjusting the budgets and periods, the end-to-end latency is reduced while keeping the loss-rate to 0. Although the total utilization increases, it is already much smaller because of scaling the period by $\alpha$. This is the reason *CoPi* succeeds in scheduling task pipelines under the constraints.

*CoPi* enforces that task runtime budgets are never decreased from $C$, as a fraction of a message is invalid. Also, increments to budgets are only in integer multiples of $C$.

**Explanation of Stage 2**  In this stage, *CoPi* first multiplies the equal period from stage 1 by a factor $\alpha$ ($> 1$) in Line 15. $\alpha$ is empirically chosen from a range of values and given as an input to *CoPi*. Stretching the period by an $\alpha$ factor helps in lowering the pipelined tasks' total utilization and keeps it under the RMS utilization bound. However, it violates the end-to-end delay constraint. So *CoPi* tries reducing the task periods, while it satisfies the other constraints.

Starting from the first producer-consumer pair, the producer's period is divided by $\beta$ ($> 1$) and the consumer's budget is multiplied by $\beta$, as long as the periods are greater than the respective task budgets (Lines 19–24). This reduces the end-to-end latency while keeping the producer-consumer pairs rate-matched for minimal data-loss.

Moving forward, *CoPi* checks whether the utilization bound test is satisfied in Line 25. If it passes that constraint, then end-to-end delay and loss-rate constraints are checked in Line 27. If all the constraints are satisfied, a schedulable pipeline with new budget and period assignments to its tasks is returned.

If a constraint is violated, the algorithm moves on to the next pair in the pipeline and repeat the steps until it covers all the producer-consumer pairs in the pipeline. After completing an iteration of going through all the producer-consumer pairs of a pipeline, *CoPi* again starts from the first pair for another iteration in Line 19. If no pair could be tuned for an iteration, tracked by the *one_pipe_changed* boolean variable, *CoPi* moves on to stage 3.

### 5.2.3  Stage 3

Lines 37–51 in Algorithm 1 show Stage 3. The objective of this stage is to further reduce the E2E delay, while keeping the total task utilization the same and the loss-rate under its upper bound.

In this stage, *CoPi* scales down the budgets and periods of the tuned consumers of stage 2. The stage starts from the sink or the last task of a pipeline and goes until the source task. It divides both periods and budgets of a task by $\beta$, as long as the budget is more than or equal to the initial budget of the task to process a single message. In each iteration, the constraints are checked in Line 48 and if they are satisfied, a feasible task

pipeline is returned. Otherwise, *CoPi* declares the pipeline unschedulable.

### 5.2.4 Discussion

*CoPi* runs Stage 1 only once for a pipeline, and it runs Stage 2 and 3 multiple times with different α values. In all our experiments, we set β to an empirically chosen value of 2, while we test Stages 2 and 3 with α in the range 1.01 to 2. Next, we establish a lower bound on α in Equation 14 to minimize the runtime overhead in Section 5.2.5. The steps of incrementing α and its higher bound can also be used to control the runtime overhead of the algorithm.

### 5.2.5 Lower Bound on α

The equal period derived in Stage 1 is $T_{eq} = \dfrac{E^{UB}}{N+1}$. Therefore, the total task pipeline utilization is $\sum_{i=1}^{N} \cdot \dfrac{C_i}{T_{eq}}$. *CoPi* moves to the second stage because this total utilization is not under the utilization bound of RMS. So, $\sum_{i=1}^{N} \dfrac{C_i}{T_{eq}} > B$, where $B = n \times (2^{\frac{1}{n}} - 1)$.

In step 1 of Stage 2, *CoPi* scales the equal period by α. Therefore, the total utilization after scaling must be less than or equal to $B$. Otherwise, *CoPi* will not be able to adjust the task budgets and periods in its next steps. Therefore:

$$\sum_{i=1}^{N} \frac{C_i}{\alpha \times T_{eq}} \le B$$

$$\sum_{i=1}^{N} \frac{C_i}{\dfrac{\alpha \times E^{UB}}{N+1}} \le B$$

$$\frac{N+1}{\alpha \times E^{UB}} \sum_{i=1}^{N} C_i \le B$$

$$\frac{N+1}{B \times E^{UB}} \sum_{i=1}^{N} C_i \le \alpha$$

$$\frac{N+1}{N \times (2^{\frac{1}{N}} - 1) \times E^{UB}} \sum_{i=1}^{N} C_i \le \alpha \ \text{(for RMS)} \qquad (14)$$

Equation 14 provides a starting value of α to *CoPi* with the above lower bound formula.

### 5.3 Examples

Figure 7 shows an example iteration of *CoPi* where it schedules a pipeline of 5 tasks. The constraints are shown at the top left. The utilization bound is the RMS bound for 5 tasks. We show a successful iteration where α is set to 1.329 (rounded down to 1.32 to 2 decimal places). The pipeline parameters are shown on the left at each stage. Their colors indicate if a parameter constraint is satisfied at a stage (green – satisfied,

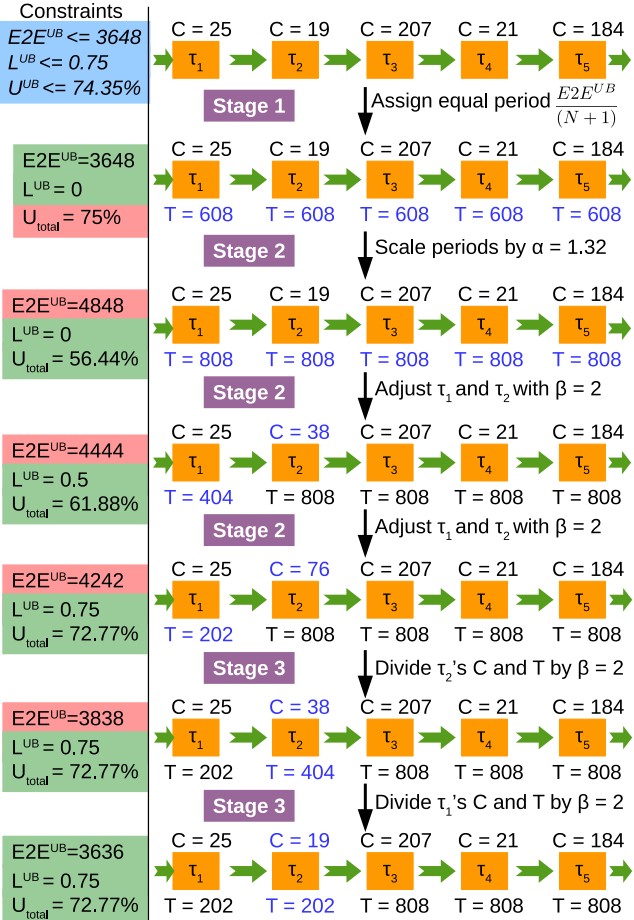

Figure 7: *CoPi* Example 1.

red – unsatisfied). Figure 8 shows another such example but with 0% loss-rate constraint, or no data loss.

## 5.4 Execution Time Complexity of *CoPi*

For a pipeline of length $N$, stage 1 runs in $O(N)$ time. Stage 2 iterates over the length of a pipeline multiple times, depending on the constraints. Stage 2 starts with a lower total pipeline utilization and goes up to the RMS utilization bound. Since it divides the task periods and multiplies the budgets by a constant β, the number of iterations in Stage 2 is some constant. It is calculated by a function dependent on logarithm base α of task budgets, e2e delay upper bound and utilization bound. As all of them are constant, stage 2 approximately takes $\frown O(K \times N)$ (where $K$ is a constant, and $N$ is the pipeline length), because it checks all the producer-consumer pairs in a pipeline.

Finally, Stage 3 checks all the tasks in a pipeline and runs in $O(N)$. Tuning each task in Stage 3 also takes a similar logarithmic function of the task budgets and can be assumed to be $\frown O(J)$, for some constant $J$. Therefore, Stages 2 and 3 together take approximately $O((K+J) \times N)$, where $O(K+J)$ represents the hardness of the constraints.

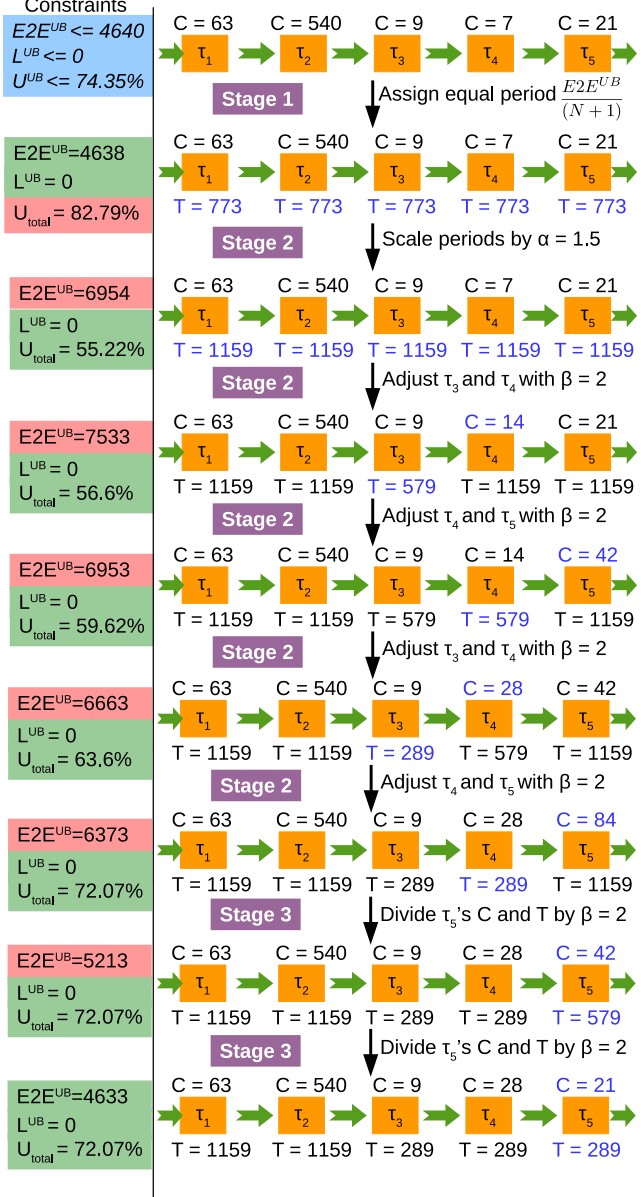

Figure 8: *CoPi* Example 2.

These stages are run for a limited range of α values. For example, if we test all α values between 1.01 and 2 with a 0.01 step increment, Stages 2 and 3 are then run for 100 times. We could also find a feasible schedule before that. So we assume the total number of Stage 2 and 3 runs to be a constant *I*. However, the lower bound of α has an inverse relationship with *N*. Therefore, the range of α values are higher for a larger *N*. So, *I* leans to $\frown O(N)$.

*CoPi* overall takes $O\big(I \times (K+J) \times N\big)$, where $O((K+J) \times I)$ is dependent on the constraints. Overall, *CoPi*'s complexity is linear to quadratic with respect to a pipeline's length. Thus, *CoPi* is a good candidate as a helper heuristic to a scheduling algorithm at runtime. Section 7.2 presents an experimental analysis of *CoPi*'s runtime overhead.

# 6 Multiprocessor Pipeline Scheduling

In multiprocessor scheduling, we utilize *CoPi*'s feature of providing a set of independent and asynchronous tasks that are chained in a pipeline. Thus, the asynchronous tasks are free to be mapped to any available processor. Even runtime task migrations are possible, as there are no data, timing or priority dependencies between tasks after *CoPi* has derived the task parameters. We use Equation 2 by Davare *et al.* to derive the upper bound on the worst-case end-to-end delay, in contrast to Equation 8 used for uniprocessor scheduling, to avoid any priority dependencies between the tasks. In this way, a task could be mapped to any available processor as long as the tasks complete their jobs within their periods. Hence, *CoPi* uses the previous two Equations 9 and 10 from Section 3, and Equation 15 given below in this case.

$$2 \times \sum_{i=1}^{N} \times T_i \leq E^{UB} \tag{15}$$

This multiprocessor pipeline scheduling algorithm demonstrates the benefits and application of the constraint solving approach and *CoPi*. *CoPi* is coupled with a traditional and well-known multiprocessor scheduling heuristic for this purpose. Although multiprocessor scheduling is an NP-hard problem [38], there are well-known heuristics to map a set of tasks to a number of processors in polynomial time. We use the worst-fit decreasing (WFD) heuristic. In the WFD heuristic, the algorithm maintains a sorted list of pipelined tasks based on their utilization values, and a sorted list of the processors based on the available utilization, both in decreasing order. Then, it maps a task to a processor, starting from the head of each sorted list. After *CoPi* provides a set of tasks, the algorithm uses WFD to map a pipeline to the available processors. In addition, the algorithm implements a few other heuristics to improve the runtime acceptance ratio of new pipelines, that are explained in Section 6.1 and 6.2.

The algorithm uses partitioned RMS scheduling for separate processors, where each processor runs its assigned tasks. The RMS utilization bound [48] is checked for task acceptance to a particular processor. The algorithm tracks each processor's available utilization by initially approximating it to 0.69 – the RMS bound for infinite number of tasks.

The multiprocessor scheduling algorithm is outlined in Algorithm 2. Auxiliary scheduling driver code externally initializes the number of processors and their utilizations. Then, the driver calls Algorithm 2 to map a pipeline with constraints to available processors. Algorithm 2 takes a pipeline and its constraints as input and returns True if the pipeline is accepted, otherwise it returns False.

Algorithm 2 works as follows: It first determines the sum of available utilizations across all processors in Line 7. This available utilization is fed to *CoPi* along with a new pipeline's task budgets and constraints in Line 8. *CoPi* either returns a

---

**Algorithm 2** Multiprocessor Pipeline Scheduling

---

1: Input: *pipeline* - Budgets of *N* Pipelined Tasks in ordered sequence from Source to Destination
2: Input: *e2e_ub* - upper bound on end-to-end delay
3: Input: *loss_ub* - upper bound on loss-rate
4: Input: α - The multiplicative scaling factor
5: Input: β - The divisive scaling factor
6: Output: True, if a pipeline is accepted in the multiprocessor system, else: False.
7: *util_ub* = `get_total_util_from_all_procs()`
8: *q_pipeline* =
     Call *CoPi*(*pipeline*[*budgets*], *e2e_ub*, *loss_ub*, *util_ub*, α, β)
9: **if** *q_pipeline* is *None* **then**
10:     Try reducing utilization of an already admitted pipeline
11:     Then go back to Line 7.
12:     Do the above for only a limited time.
13: **end if**
14: *iter* = 0
15: **while** *iter* ≤ *num_core* **do**
16:     **if** `WFD_FIT`(*q_pipeline*) is not successful **then**
17:         //try migration
18:         Sort the processors in decreasing order of available utilization
19:         **for** each processor *p* in the above sorted list **do**
20:             Sort the mapped tasks in decreasing order of utilization assigned to *p*
21:             **for** each task *t* in the above sorted list **do**
22:                 **for** each processor *q* among all processors **do**
23:                     **if** *t*'s utilization ≤ *q*'s available utilization and *p* ≠ *q* **then**
24:                         unmap *t* from *p* and map to *q*
25:                         break out of for loop at Line 19
26:                     **end if**
27:                 **end for**
28:             **end for**
29:         **end for**
30:         Increment *iter*
31:     **else**
32:         **return** True
33:     **end if**
34: **end while**
35: **return** False

---

pipeline with schedulable budget and period assignments, or returns `None` if the pipeline was unschedulable. For a schedulable pipeline, the algorithm tries mapping individual tasks to processors. A task is able to be mapped to any available processor, since *CoPi* generates independent tasks.

Then, in Line 16, the algorithm uses a `WFD_FIT` function for the WFD heuristic, to map new pipelined tasks to processors. It first determines whether all the pipelined tasks could be mapped, by checking the individual tasks one by one, before actually mapping the tasks to processors. Only if it could map all the tasks in a pipeline to the available processors, are they actually assigned to local scheduling queues.

If the WFD heuristic is not able to map all the tasks, the algorithm tries migrating tasks from one processor to another in Lines 17– 29. The migration strategy is described later in Section 6.1.

Moreover, when *CoPi* is first called in Line 8, it may not return a feasible schedule because of not meeting the utilization bound. Such an infeasible schedule may occur due to unoptimized pipelines already admitted in the system. As the system starts with more available processor utilization, *CoPi*

is initially run with a higher and relaxed utilization bound constraint. Hence, it may have returned unoptimized pipelines because they were already schedulable with higher utilizations. In such cases, the multiprocessor scheduling algorithm attempts to reduce the utilization of an admitted pipeline in Lines 10–12. The strategy is explained further in Section 6.2.

## 6.1  Runtime Task Migration

When the multiprocessor scheduling algorithm fails to schedule a new pipeline on its available processors even after having spare utilization, it explores the possibility of migrating already mapped tasks to make room for a new pipeline. Lines 17– 29 show how this is done in Algorithm 2. The algorithm first sorts the processors in decreasing order of available utilization. For each processor in the sorted list, it sorts the mapped tasks in decreasing order of task utilization. It picks a task from this sorted list of mapped tasks, and migrates it to the first available processor that can accommodate the task.

As soon as a task is migrated, the algorithm tries to schedule the new pipeline using the WFD heuristic. We do this to minimize the number of total task migrations in the system, because migrations have practical runtime overhead. For a new pipeline, the algorithm only tries migrating *M* tasks at most, where *M* is the number of total processors. Thus, we limit the number of migration attempts per new pipeline to bound the time to find a schedulable mapping.

In summary, we employ task migration to admit more pipelines at runtime by creating larger utilization holes in processors. However, task migration should be carefully administered and minimized as it is associated with non-negligible overhead and potential disruptions for admitted pipelines. Nevertheless, predictable migration [47] enables admission of new pipelines in a multiprocessor system. Our evaluation results in Section 7.4 demonstrates the benefit of migrations in terms of the number of dynamically accepted pipelines.

## 6.2  Runtime Pipeline Optimization (RPO)

We provide another enhancement for multiprocessor scheduling by attempting to reduce the total utilization of an already admitted pipeline. When *CoPi* fails to schedule a new pipeline, Algorithm 2 picks an already admitted pipeline and calls *CoPi* with a tighter utilization bound. For our experiments, Algorithm 2 asks *CoPi* to reduce a pipeline's current utilization by 5% at a time. If *CoPi* is able to find new task budgets and periods for the admitted pipeline with the new utilization bound constraint, the algorithm unmaps all the tasks of the pipeline from corresponding processors. It then remaps tasks with new parameters following the WFD heuristic. This strategy reduces the total processor utilization at runtime and also makes room for a new pipeline. Our evaluation shows that this strategy yields a higher number of

pipeline admissions. However, the *RPO* implementation in a working system should ensure that the task unmapping and reassignment are done at a safe timing point. If a task $\tau_i$ is running, then the scheduler may wait $T_i$ time before new runtime and budget can be applied. Therefore, the scheduler needs to wait $(\sum_{i=1}^{N} T_i)$ time units in the worst-case. An RPO implementation in an RTOS needs to be aware of such delays. The full analysis and implementation details of RPO are out of scope of the paper and left for future work. In this paper, we show the benefits of *RPO* in admitting new pipelines at runtime with simulated experiments in Section 7.4.

## 7 Evaluation

We evaluate *CoPi* and the multiprocessor scheduling algorithm on top of it by running simulated experiments [3]. We first generate individual task utilizations using the standard UUnifast algorithm [8]. Then, we generate the task budgets by multiplying the utilization with a random value chosen from uniform distribution between 100 and 1000. These values are used as the initial task runtime budgets, $C$ in the model. The constraint solvers, including *CoPi*, use the task budgets to solve the E2E delay, loss-rate and utilization bound constraints.

Before going into the analysis of evaluation results, we explain several parameters related to the E2E delay, to standardize its relationship to the task budgets. These parameters are used throughout this section.

1. **Latency Budget Gap (LBG):** This is the ratio of a supplied upper bound on the E2E latency ($E^{UB}$) and the summation of all the task budgets in a pipeline $P$.

$$\text{Latency Budget Gap: } LBG = \frac{E^{UB}}{\sum_{i=1}^{N} C_i} \quad (16)$$

LBG depicts the gap, or surplus time, beyond the sum of task budgets to the E2E delay. As noted in Equation 2 and 4, the periods contribute to the E2E delay. Therefore, LBG intuitively depicts the hardness of the E2E delay constraint.

For different pipelines, the task budgets are different. Therefore, their end-to-end delay upper bounds ($E^{UB}$) are also expected to be different. This makes it difficult to compare how a solver performs for different pipelines with different task budgets for the end-to-end delay constraint. LBG standardizes the relationship between the task budgets and $E^{UB}$. Thus, performance against randomly generated task pipelines are compared for different solvers with different values of LBGs.

---

[3]The artifacts of the experiments are available at https://github.com/sohamm17/pipe_schedule

A higher LBG means that the upper bound on the E2E delay is greater, and the E2E delay constraint is more relaxed. Finding a schedulable pipeline is more probable with a higher LBG, because of the potential for a greater number of possible task constraint solutions. Conversely, a smaller LBG means a tighter E2E delay constraint, as the gap between $E^{UB}$ and the sum of task budgets is reduced.

2. **Normalized LBG (NLBG):** This is the ratio of the LBG and the length of a pipeline. NLBG normalizes LBG with respect to the pipeline length.

$$\text{Normalized LBG: } NLBG = \frac{LBG}{N} \quad (17)$$

We can only compare the pipelines of same length with LBG. With NLBG, pipelines of different lengths are compared (e.g., Figure 11). A higher NLBG, as with LBG, also increases the probability of finding a schedulable pipeline, and vice-versa.

The loss-rate is already expressed in terms of percentage, so we do not need any other standardized parameter for it.

We run all the experiments with Python 3.6 on an 64-bit Linux (Ubuntu 16.04) machine featuring a Core i5-4210 processor. For every experiment, we report the average value against 1000 randomly generated task pipelines. We choose $\beta = 2$ in all cases, whereas $\alpha$ is iterated starting from 2 and then decreasing in steps of 0.01.

### 7.1 Uniprocessor Acceptance Ratio

#### 7.1.1 Only End-to-end Delay Constraint

The first experiment compares the pipeline acceptance ratios (ARs) of open-source constraint solvers to *CoPi*'s AR, only under the end-to-end delay constraint. Figure 9a shows the percentage of pipelines with 10 tasks that are schedulable for a uniprocessor RMS utilization bound against increasing LBG. LBG is varied from an acceptance ratio of 0% to 100% for most solvers.

The GEKKO Optimization Suite [5] with its APOPT solver [1] dominates other modeling packages and *CoPi*. Although pyomo [11,34] uses a well-known IPOPT method [67] for MINLP problems, its implementation of the *Disjunction* properties are still in development [54]. Hence, its acceptance ratio is worse, but the performance improves with higher *LBG* values. scipy is a more generalized mathematical and optimization Python package, from which we use the *trust-constr* [12, 46] constraint minimization approach. As it is a local minimizer, its solution is dependent on the initial suggested value. Because of limitation in its current implementation [61], it performs poorly for the same initial value that is provided to GEKKO and pyomo.

*CoPi*'s AR is worse than GEKKO's, but better than other solvers. It reaches 100% AR at *LBG* = 16 when GEKKO

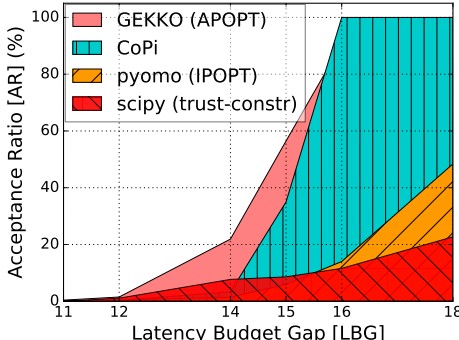

(a) Solvers under an E2E delay constraint.

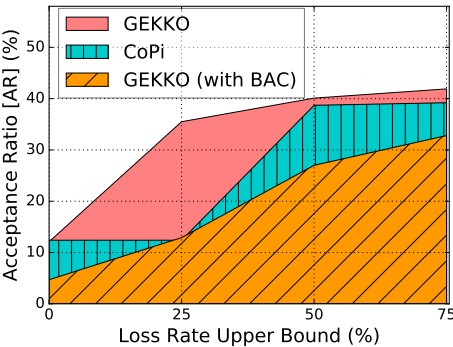

(b) Solvers under both pipeline constraints (*LBG* = 15).

Figure 9: Uniprocessor pipeline acceptance ratios (*N* = 10).

also reaches near 100% AR. Its performance is similar for smaller ($\leq$ 12) and larger ($\geq$ 15) *LBG* values. This experiment shows that *CoPi* performs comparably with respect to the other MINLP solvers. The performance of the MINLP solvers may well be further improved with more iterations, commercial solvers and perhaps better modeling techniques, but MINLP solvers are not suitable for runtime scheduling because of their slow execution time performances. We show this in Section 7.2.

### 7.1.2   Combined E2E Delay and Loss-rate Constraints

We now focus on the performance of GEKKO and *CoPi*, as the other solvers do not perform as good as these two. For the next set of experiments, we apply both E2E delay and loss-rate constraints to task pipelines. Apart from running GEKKO with the existing constraints, we also run it with the Budget Adjustment Constraint (BAC) (described in Section 4.1) to investigate whether it is able to utilize a rate-matching domain knowledge. AR is plotted in Figure 9b for increasing loss-rate upper bounds against a fixed *LBG* = 15 and *N* = 10.

The graph shows that *CoPi* performs comparably to GEKKO. As the upper bound on the loss-rate increases, AR also improves for both the solvers. GEKKO (with BAC) performs worst because BAC adds more variables to the solver. It exhausts the number of iterations with more variables. GEKKO performs much better without BAC, although is not able to exploit a rate-matching heuristic like *CoPi* does.

## 7.2   Solver Runtime Overhead

In the above experiments, we show that *CoPi* and GEKKO perform similarly against the E2E delay, loss-rate and RMS utilization bound constraints. In the next experiments, we investigate the execution times of both solvers, to examine their capabilities in runtime scheduling of task pipelines.

Figure 10a plots the runtime of *CoPi* and GEKKO for schedulable pipelines against an increasing LBG for pipeline length of 10. We only plot for *LBG* = [13, 15] because the acceptance ratio is significant in this range for both GEKKO and *CoPi*. It shows that the runtime is comparable for both the solvers for a stricter LBG. As LBG increases, the E2E delay constraint is more relaxed. In these cases, *CoPi* is able to find a schedulable pipeline more quickly than GEKKO is capable of doing. For *LBG* = 15, GEKKO takes on average almost 5 times more than *CoPi*.

Figure 10b plots the execution times for unschedulable pipelines. It shows that even for an unschedulable pipeline, GEKKO keeps searching for feasible task parameters for a longer time before retiring, whereas *CoPi* responds *at least* 2-3 times faster.

Figure 10c shows runtime overhead of *CoPi* and GEKKO with respect to increasing pipeline length. As explained in Section 5.4 for *CoPi*, its runtime increases with increasing length of a pipeline (against a fixed *NLBG* = 1.5). The relationship between runtime and pipeline length is nearly linear. GEKKO's runtimes for both schedulable and unschedulable pipelines are greater than *CoPi*'s runtimes for all pipeline lengths. More importantly, GEKKO's runtimes grow faster with pipeline length than *CoPi*'s do.

In the next experiment, we evaluate the effect of E2E delay and loss-rate constraints on the runtime overhead for GEKKO and *CoPi*. Table 2 summarizes the result of each experiment with a pipeline length of 10 and two constraints (*LBG* = 15 fixed , $L^{UB}$ varied). *CoPi* always has lower runtime overhead compared to GEKKO. The performance of GEKKO degrades significantly after adding both the constraints for failed pipelines from 327ms to as much as 2237ms, whereas *CoPi* takes similar time to fail to schedule a pipeline. The reason is that *CoPi* checks the loss-rate constraint every time where the E2E delay constraint and utilization bound are checked. Hence, the failing time does not increase. However, the runtimes for schedulable pipelines do increase for *CoPi* after adding the loss-rate constraint on top of E2E delay, because it discards all the results where loss-rate is greater than the given upper bound.

Table 2: Runtime overhead for both pipeline constraints (*N* = 10).

| Constraints | Accepted (ms) | | Failed (ms) | |
|---|---|---|---|---|
| | GEKKO | *CoPi* | GEKKO | *CoPi* |
| E2E Delay (*LBG* = 15) | 61 | **12** | 327 | **128** |
| E2E + Loss-rate ($L^{UB} \leq 0\%$) | 205 | **104** | 966 | **130** |
| E2E + Loss-rate ($L^{UB} \leq 25\%$) | 191 | **105** | 1437 | **132** |
| E2E + Loss-rate ($L^{UB} \leq 50\%$) | 187 | **102** | 1801 | **135** |
| E2E + Loss-rate ($L^{UB} \leq 75\%$) | 200 | **107** | 2237 | **131** |

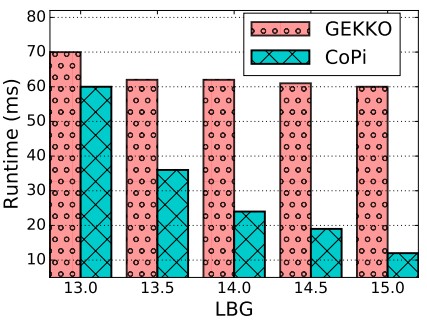

(a) Runtime for schedulable pipelines ($N = 10$).

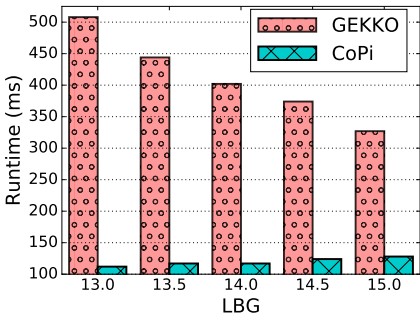

(b) Runtime for unschedulable pipelines ($N = 10$).

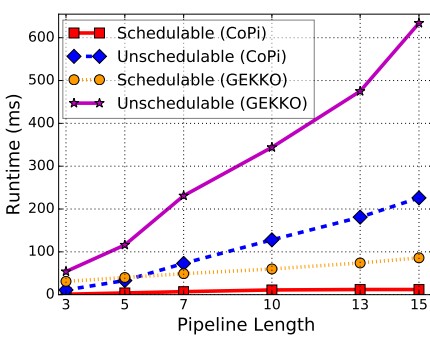

(c) Runtime vs. pipeline length ($NLBG = 1.5$).

Figure 10: Runtime overhead under an end-to-end delay constraint.

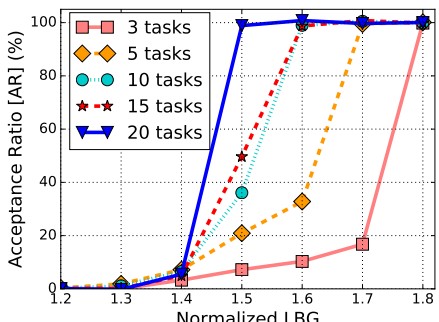

Figure 11: NLBG against different pipeline lengths.

## 7.3 Performance Insights of *CoPi*

In this section, we delve into more details about *CoPi*'s performance and its optimization techniques.

### 7.3.1 Pipeline Length and *NLBG*

Figure 11 shows *CoPi*'s AR with increasing NLBG for different pipeline lengths only under the E2E delay constraint. After certain NLBGs, the acceptance ratio jumps to 100% for all pipeline lengths because: 1) assigning fixed and equal periods of $\frac{E^{UB}}{N+1}$ meets the E2E delay constraint requirement, and 2) as individual task utilizations reduce with greater periods, the utilization bound constraint is also satisfied.

The NLBG value, after which most pipelines are schedulable, is dependent on the pipeline length. For example, Figure 11 shows that all pipelines are schedulable for pipeline lengths of 20 for $NLBG \geq 1.5$. However, for pipeline lengths of 5, all pipelines are schedulable for $NLBG \geq 1.7$. This threshold NLBG is higher for shorter pipelines because there are fewer available pipelined tasks to tune. *CoPi* gets fewer opportunities to distribute the periods from the E2E delay, before the utilization bound constraint is violated.

### 7.3.2 Effectiveness of Stage 2 and 3

With smaller and stricter NLBG, it is harder to find a schedulable pipeline. Table 3a shows the AR of *CoPi*'s Stages 2 and 3

with varying pipeline length and strict NLBG values. Together with Figure 11, this demonstrates that *CoPi*'s Stages 2 and 3 are able to find more feasible pipelines for stricter NLBGs for any pipeline length. For example, *CoPi*'s Stages 2–3 AR is 31.8% for $NLBG = 1.6$ and $N = 5$, whereas *CoPi*'s overall AR is also 31.8% for the same pipeline length and NLBG (see Figure 11). This shows that Stages 2 and 3 contributed to all the pipeline acceptances for these constraints. Moreover, *CoPi*'s optimizations are more useful for longer pipelines because Stages 2 and 3 are able to tune more tasks in a pipeline to satisfy the constraints. It can be seen that Stages 2 and 3 schedule as many as 49% of pipelines ($N = 15, NLBG = 1.6$) in Table 3a.

### 7.3.3 Utilization Bound

*CoPi* checks the Liu-Layland RMS utilization bound to determine whether a pipeline is schedulable on a uniprocessor. However, the RMS utilization could be relaxed to 1 if all the tasks are harmonic [44]. Exploiting this RMS scheduling property, Table 3b shows that *CoPi* is able to schedule more tasks, even with very strict NLBGs.

Table 3: *CoPi*'s performance insights.

(a) Stage 2 and 3 acceptance ratio under *tight* NLBG.

| Pipeline Length | NLBG | AR (%) |
|---|---|---|
| 3 | 1.3 | 0.8 |
| | 1.4 | 2.2 |
| | 1.5 | 7.4 |
| | 1.6 | 11.1 |
| 5 | 1.3 | 2.1 |
| | 1.4 | 6.5 |
| | 1.5 | 22 |
| | 1.6 | 31.8 |
| 10 | 1.3 | 2.5 |
| | 1.4 | 6.7 |
| | 1.5 | 7.2 |
| | 1.6 | 35.5 |
| 15 | 1.3 | 1.1 |
| | 1.4 | 1.7 |
| | 1.5 | 4.8 |
| | 1.6 | 49 |

(b) Acceptance ratio improvement with larger utilization bound for harmonic tasksets ($N = 10$).

| NLBG | RMS Utilization Bound | AR (%) |
|---|---|---|
| 1.1 | $\leq n \times (2^{\frac{1}{n}} - 1)$ | 0 |
| | $\leq 1$ | 20.4 |
| 1.2 | $\leq n \times (2^{\frac{1}{n}} - 1)$ | 0.3 |
| | $\leq 1$ | 67.6 |
| 1.4 | $\leq n \times (2^{\frac{1}{n}} - 1)$ | 3.3 |
| | $\leq 1$ | 98.7 |
| 1.5 | $\leq n \times (2^{\frac{1}{n}} - 1)$ | 35 |
| | $\leq 1$ | 100 |
| 1.6 | $\leq n \times (2^{\frac{1}{n}} - 1)$ | 100 |
| | $\leq 1$ | 100 |

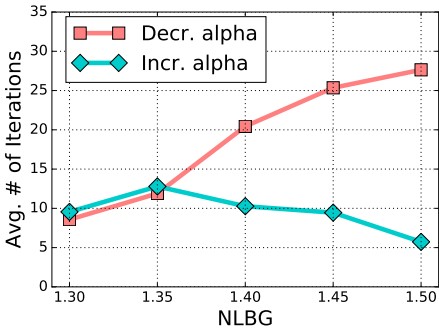

Figure 12: Number of iterations in *CoPi* against α iteration strategies.

### 7.3.4 Other Variations

In this set of experiments, we compare two strategies of iterating over α within a fixed range: incrementing and decrementing. We investigate which one reduces the number of *CoPi* optimization loop iterations of Stages 2 and 3 combined. For α's increment, we start from the value derived by Equation 14 and increase until α = 2 to limit the number of iterations. For decrement, we start from a higher α value (2 in the experiment) and decrease until 1.01, or the point when scaling up periods by α does not meet the utilization bound test.

Figure 12 shows the number of iterations for schedulable pipelines of length 10. As *NLBG* increases from 1.3 to 1.5, incrementing α takes a smaller numbers of iterations to find feasible period assignments. Decrementing α is thus not preferable. Both of these techniques have similar acceptance ratio.

## 7.4 Multiprocessor Performance

In this section, we investigate the performance of our multiprocessor scheduling algorithm coupled with *CoPi*. We measure the number of pipeline acceptances for dynamically appearing pipelines in a simulated environment that models a runtime scheduling scenario.

We experiment with 2, 4 and 8 processors. We feed 50, 100, and 200 pipelines respectively to the 2, 4, and 8 processors. For each of these experiments, pipelines are fed one after another to simulate dynamic arrivals. After every 5, 10 and 20 pipelines respectively for 2, 4 and 8 processors, all the pipelines are unmapped from the processors to simulate ephemeral pipelines. We also vary the pipeline length.

Figure 13 shows the number of accepted pipelines for different pipeline lengths and number of processors. Here, WFD stands for the worst-fit decreasing heuristic, *mig* is runtime task migration (described in Section 6.1), RPO is runtime pipeline optimization (described in Section 6.2). We show the performance for WFD without migration and RPO, WFD with only RPO (+*RPO only*) and migration (+*mig only*), and with both of them together (+*both*). This experiment reveals that combining WFD with runtime optimizations accommodates more dynamic pipeline arrivals.

(WFD+*RPO only*) has limited ability to accept more pipelines in a number of cases ((N=3 and 10, M=2), (N=3 and 10, M=4), (N=all, M=8), where M is number of processors). Due to the limitations of the WFD heuristic, RPO alone cannot help significantly in accommodating more pipelines, unless already admitted tasks create utilization holes for new pipelines after optimization. Nevertheless, applying both runtime strategies together (+*both*) results in more pipeline admissions for all processor combinations and pipeline lengths. After task migrations create larger available processor utilizations, RPO helps to accommodate more pipelines.

In Figure 13a, both migration and RPO are not able to accommodate more pipelines for N = 10 using 2 processors. As the pipeline length is longer, the algorithm cannot accommodate all the tasks of a single pipeline using just 2 processors. Hence, the number of pipeline admissions does not improve.

### 7.4.1 Processor Utilization

Table 4 tabulates the normalized (per processor) utilization at the end of the experiment for varying pipeline lengths. It shows that RPO indeed decreases the processor utilization on average. However, it does not adjust the *available utilization holes* in processors for new tasks. Adding migration with RPO improves the per processor utilization in addition to admitting more pipelines.

The single-CPU RMS utilization bound is 69%, when the number of tasks tends to infinity. The rest of the CPU is usually given to lower-priority background tasks. Our multiprocessor algorithm actually keeps the processor utilization to a respectably high level, as displayed in Table 4.

### 7.4.2 Task Migrations

Table 5 shows the average number of task migrations which resulted in the successful scheduling of new pipelines. The number of migrations increases with more processors, as the algorithm limits the number of migrations per new candidate pipeline to the number of available processors. Overall, this experiment shows that only a few migrations are needed to accommodate new pipelines in the system, and the average numbers of migrations are much smaller than their limits.

Table 4: Multiprocessor utilization.

| Pipeline Length | Strategy | Normalized Utilization Per Processor (%) |
|---|---|---|
| 3 | WFD | 51.2 |
| | + RPO only | 50.5 |
| | + migration only | 54.8 |
| | + both | 54.2 |
| 5 | WFD | 54.9 |
| | + RPO only | 55 |
| | + migration only | 60.8 |
| | + both | 62.46 |
| 10 | WFD | 61.6 |
| | + RPO only | 60.8 |
| | + migration only | 65 |
| | + both | 64.4 |

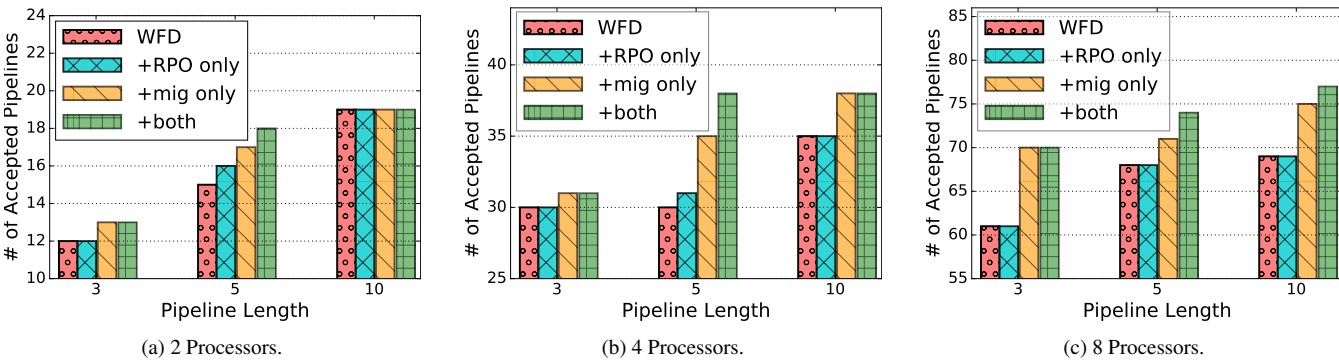

(a) 2 Processors.   (b) 4 Processors.   (c) 8 Processors.

Figure 13: Number of accepted (schedulable) pipelines in multiprocessors.

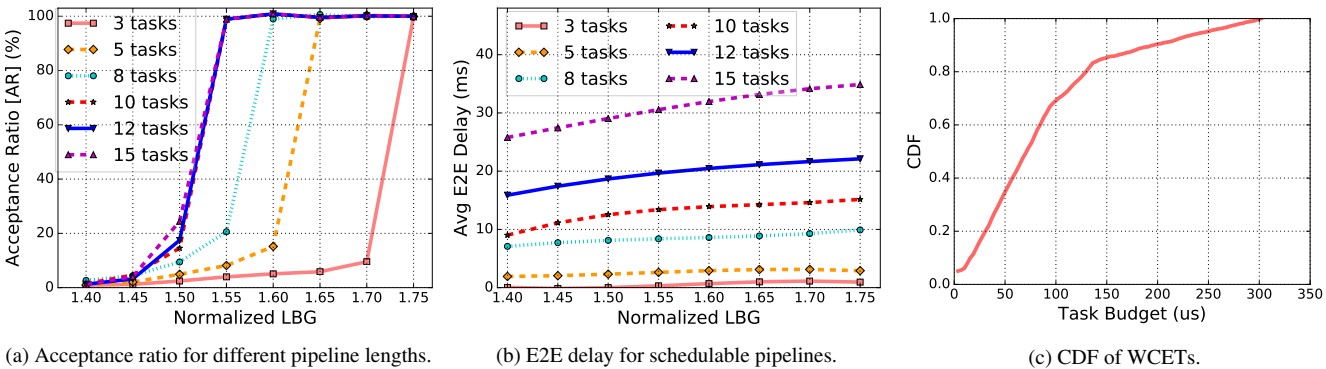

(a) Acceptance ratio for different pipeline lengths.   (b) E2E delay for schedulable pipelines.   (c) CDF of WCETs.

Figure 14: Experiments with dataset from the WATERS 2015 workshop paper [43].

Table 5: Task Migrations.

| Processors | Strategies | Average Migrations |
|---|---|---|
| 2 | WFD + migration only | 0.1 |
| | WFD + migration + RPO | 0.13 |
| 4 | WFD + migration only | 0.67 |
| | WFD + migration + RPO | 0.67 |
| 8 | WFD + migration only | 2.7 |
| | WFD + migration + RPO | 3.17 |

## 7.5 Experiments with an Industry Benchmark

We have tested *CoPi* with a benchmark from the WATERS 2015 workshop paper [43], provided by Bosch, for uniprocessor scheduling. We calculate the worst-case execution time of a task by multiplying a random average case execution time (ACET) with a random WCET factor. The random ACET and WCET factor are chosen from the task distribution provided in Table III of the paper. We consider pipeline lengths from 3 to 15 because each runnable entity from the dataset might be re-implemented as a pipelined task.

Figure 14 summarizes the result of this experiment. Figure 14a shows the acceptance ratio among 1000 randomly generated tasksets for each case. The graph is similar to Figure 11, but focuses on NLBG between 1.4 to 1.7 for more fine-grained data points. Figure 14b captures the average E2E delay for schedulable pipelines in ms, where each runnable's WCET is in $\mu$s granularity. Figure 14c shows a CDF of the worst-case execution times for 3 tasks pipelines. Other

pipeline lengths have similar CDF. In Figure 14b, the E2E delays for smaller pipelines are under 10ms which is usually expected in automotive industries. The E2E delays for longer pipelines go up to 30 ms depending on the NLBG. As longer pipelines might be utilized for comparatively lower frequency workloads, slightly longer E2E delays are tolerable.

## 8 Discussion on Implementation

In this section, we discuss some future implementation opportunities of *CoPi*. Most RTOSs, including FreeRTOS [25] and Quest [18], support RMS and other static priority preemptive uniprocessor scheduling algorithms. As these RTOSs are increasingly adopted in sophisticated automotive and industrial domains [26, 63], guaranteeing task pipelines will be one of the crucial features to support. These applications are highly dynamic in nature, with task pipelines that arrive and terminate at runtime [30, 37]. The uniprocessor pipeline scheduling algorithm with *CoPi* can be integrated in these OSes. Since it is difficult to implement a complete MINLP solver in an OS scheduler, the usage of heuristics like *CoPi* is appealing. By integrating *CoPi*, operating systems will be able to provide end-to-end guarantees in task pipeline scheduling.

Moreover, micro-ROS [52], a lighter implementation of ROS [57], is now supported on a few RTOS such as FreeRTOS [24] and QNX [56]. As ROS tasks and services are

already modeled as asynchronous tasks communicating over a publisher-subscriber model, a ROS task pipeline is similar to that described in this paper. Therefore, *CoPi*'s implementation in an RTOS will be useful in providing end-to-end guarantees in ROS cause-effect chains. *CoPi*'s benefit to multiprocessor scheduling is also applicable to RTOSs that provide SMP support [18, 27, 55]. Nevertheless, further studies are needed before support for real-time ROS task pipelines is implemented in practical SMP systems [13, 65].

## 9   Related Work

Many past researchers have studied scheduling algorithms for tasks with data dependencies. Some have proposed using a DAG to express task-level data dependencies and precedence constraints as part of multicore scheduling [15, 35, 45, 59, 66, 69, 71]. Our work focuses on the use of end-to-end service constraints to establish scheduling parameters for pipelined tasks executing on uni- and multiprocessor platforms.

Gerber et al. presented a generic framework that shows how constraint programming helps in guaranteeing end-to-end constraints in a task graph [28]. Davare *et al.* presented an end-to-end timing analysis of task pipelines and provided an upper bound on the end-to-end latency [19]. This work is closely related to our work, as it also proposes the problem of finding task periods as an optimization problem under constraints, which is solved by geometric programming. However, the work does not consider loss-rate as a constraint and instead focuses on latency. In this work, we introduce loss-rate as one of the constraints and refine the optimization problem to find suitable task periods and budgets. We use an improved end-to-end latency analysis by Dürr *et al.* [20], along with the latest open-source MINLP solvers [5, 11] for comparison. These were unavailable at the time of Davare *et al.*'s work.

There are other research studies that explore the end-to-end timing analysis of task chains [6, 32, 40, 41] in practical scenarios such as in drones [16], and in ROS [13, 65]. Proposed scheduling algorithms based on these approaches rely on job release times [7, 17].

For multiprocessors, Liu and Anderson have analyzed a global scheduling algorithm for pipelined periodic tasks [49, 50]. Nevertheless, they do not consider end-to-end constraints. Finally, period selection is a widely studied problem in real-time systems [9, 14, 53, 70], even though it is not targeted specifically at real-time task pipelines.

## 10   Conclusions and Future Work

This paper explores the real-time task pipeline model and presents a non-linear optimization problem to find suitable task budgets and periods under a pipeline's end-to-end constraints and utilization bound. We propose *CoPi*, a heuristic constraint solver algorithm, which tunes task periods and bud-

gets to minimize a pipeline's loss-rate and end-to-end delay. It essentially converts pipelined real-time tasks into independent periodic tasks. We explain *CoPi* with examples, and evaluate its performance with simulations. Evaluation results show that *CoPi* performs favorably in terms of task pipeline acceptance ratio, compared to open-source MINLP solvers like GEKKO [5]. *CoPi* has an order of magnitude better runtime than GEKKO. Therefore, *CoPi* is better suited to OS-level scheduling, where implementing a relatively slow MINLP solver into a runtime system is problematic. We also demonstrate the benefits of *CoPi* in multiprocessor scheduling for dynamically arriving pipelines with fewer task migrations and more pipeline admissions.

In future work, we will explore other constraints such as throughput, for general task graphs combining multiple inputs and outputs. We have plans to deploy *CoPi* in a multicore operating system. Application frameworks such as ROS [57] and micro-ROS [52] can take advantage of *CoPi*'s support for multiprocessor scheduling, to parameterize and predictably schedule a pipeline of periodic tasks. Finally, a programming model for task pipelines is also being studied, in the context of modeling environments such as MATLAB/Simulink.

## 11   Acknowledgements

This work is funded in part by the National Science Foundation (NSF) under Grants 2007707 and 2151021. Any opinions, findings, and conclusions or recommendations expressed in this material are those of the author(s) and do not necessarily reflect the views of the NSF.

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
