# OpenReview forum: "Solution: End-to-end Scheduling of Real-time Task Pipelines on Multiprocessors"
_JSYS/2021/Nov_Papers — Submitted to JSYS Nov 21_

### Official Review · Reviewer_24AD · 2021-11-26
**Unmotivated contribution with significant problems**

**Decision:**

Strong reject: this paper has serious problems, fixing it would definitely take more than three months

**Review:**

*** Summary ***

The paper presents the formulation and solution of a mixed integer non linear programming problem to determine the best periods for pipelines of tasks that share data with one another.

*** Overall impression ***

The paper is certainly not ready for publication:

1) the relevance of the problem is not well explained,
2) there is a lack of connection to any practical application,
3) the writing and presentation style is not up to standard.

While the contribution is clear, I think it is certainly not enough for an invitation to resubmit.

*** General comments ***

I believe the paper does not really succeed in showing the relevance of the problem that is solved: periods of tasks are usually selected based on characteristics of the tasks (e.g., in cyber-physical systems sensor periods, control periods, and actuation periods are selected based on the physics of the process to be controlled, rather than being optimisation variables that can be selected at will). Given that, I wonder what is an example of a real system where the period selection technique is useful.

In terms of positioning and relevance, I would recommend taking a look at papers like "Time-selective data fusion forin-network processing in ad hocwireless sensor networks" (by Jaanus Kaugerand, Johannes Ehala, Leo Motus, and Jurgo-Soren Preden) and trying to model the sensor fusion algorithms presented in these papers with a pipeline. I do understand that the problem solved is not the same and not necessarily limited to cyber-physical systems, but with the lack of an application example in the paper, this is what comes to mind. In the paper, the authors present a sensing pipeline that can in principle be implemented with a chain (where sensor 1 passes data to sensor 2 and eventually sensor N passes data to a fusion algorithm). This will impose requirements on the end-to-end latency, but unfortunately it also imposes constraints and requirements on the period values.

Another very general comment is the following: why splitting tasks makes more sense than having a single task that invokes in sequence the functions that do $\tau_1, \dots, \tau_N$? It seems like there is a point in merging the tasks in a single big task that is actually trying to complete the pipeline in sequence within the pipeline deadline/period. No task is being used by multiple pipelines so there is no task that cannot be merged without problems.

*** Detailed comments ***

The writing in the entire paper is quite poor.

For example, the very first sentence of Section 2.1, i.e., "A task $\tau$ is a two-tuple $(C,T)$ and asynchronous.", is (already) problematic. What is a two-tuple? I suppose this means that a task is a tuple with two elements. The same appears later with a pipeline being a "four-tuple".

This said, there is an enormous confusion between derived characteristics and intrinsic characteristics. A pipeline is not a tuple with four elements. The pipeline, as modeled in the paper, is completely described by a single element: a set of tasks (each task having its own characteristics). In fact, the cardinality of the set is $N$, hence $N$ is not part of the definition of the pipeline. Even more important, given what is written in the paper $E$ and $L$ are fully defined once $S$ is known, and hence they do not belong to the definition of the pipeline.

The scheduling model also presents a set of problems. The sporadic server abstraction (probably better described in "An Application-Level Implementation of the Sporadic Server" than in [48], both being technical reports) is NOT a rate-monotonic scheduling algorithm. The two objects are quite different and they are not related in the way the authors describe. In a sporadic server, the priority of the task is dynamic and decreased when the task has completed its budget, hence this cannot be a rate-monotonic scheduling algorithm. I do understand what the authors mean with the beginning of 2.3, but this needs to be fixed. I also have a hard time seeing how using a sporadic server for periodic tasks would be the best choice (rather than e.g., a constant bandwidth server). The RMS bound that is enforced also comes as a surprise as it it clearly too conservative given the model of computation that is chosen.

At the end of 2.3, the paper reads "Moreover, the fixed execution strategy tightly bounds a pipeline's end-to-end latency [26]." - the formulation is [26] includes initial release offsets for the tasks, that are not considered in the current paper. There is therefore an inconsistency between the results taken from the literature and the model used in the paper.

This is also true for Equation (1). Paper [20] is the source of the equation. However, Theorem 5.4 in [20] specifies that the quantity bounded is the reaction time and not the worst-case end-to-end delay (as used in the paper). This is different and the term $T_1$ is therefore not needed (it refers to the fact that an event may happen precisely at the time of the start of $\tau_1$ and not be detected at the start, hence the event would only be detected and correctly reacted upon with the following job of $\tau_1$).

I also don't entirely follow the argument that leads to the transition from Equation (1) to Equation (2), which I think is quite poorly written. I do understand the point, and the fact that the quantity on the right of equation (2) is now bigger than the right-hand-side of equation (1), and therefore this is still a valid upper bound, but this needs to be expressed quite a lot better than it currently is. Furthermore, it would be good to add the (possibly acceptable) assumption that the worst case response time of each task is less than the period in the definition of the task characteristics. I think this assumption, while acceptable, is very limiting.

Another source of problems is the lack of the specification of a communication model. The results in [20], for example, are derived for an implicit communication model (see the paragraph "Communication Semantics" just before Section 2.2 in the referred paper). In the absence of a communication model between tasks, it is quite hard to understand the results and their relevance.

The definition in Equation (3) is completely unacceptable. While I understand what you mean with the sentence "total lost output messages corresponding to the input messages" and "total input messages to a pipeline", these two quantities do not mean anything if not properly defined. In particular, to calculate the loss-rate there is a need to specify a time interval and the loss rate will necessarily be a function of the time interval. Section 3.2 does not resolve how to calculate the end-to-end loss-rate of a pipeline because it does not specify the time interval.

What is calculated in Section 3.2.1 as the loss-rate is necessarily an average loss-rate over the entire execution of the tasks, but it is also important to understand and evaluate *which* events/messages are lost and which events are not.

The sentence "Therefore, the oversampling ratio: [...]" lacks a verb. The statement that when the producer has a larger period than the consumer there is no lost message in the pair is very obvious, as it is all the content of Section 3.2.1 in general. Furthermore, the statements in the section are heavily dependent on the hypotesis that the initial release offset of tasks is zero for all the tasks, while the results used from the literature do not make that assumption. This is very unconvincing.

In figure 3, the ratio between the period of $\tau_2$ and the period of $\tau_3$ is incorrect. If the period of task $\tau_3$ is 50, for every output of task $\tau_2$ (every 200) there should be 4 outputs of $\tau_3$, so we should see "1. B" four times, and then eventually "3. D" four times. Similarly, in the text "As $\tau_3$ runs twice frequently than $\tau_2$" should be corrected.

More in general, pages 4 and 5 can be significantly compressed as the statements (rules and examples) are very obvious. Similarly, the description of the algorithm is too verbose for something that is very simple.

Page 5: "A task's runtime budget $C$ to read, process and write 1 message is usually determined by profiling its worst-case execution time (WCET)" - the use of the word "profiling" here is very misleading. The WCET is usually an upper bound on the execution time of a task and is obtained using techniques like static analysis. If one obtains it via profiling, it is not really certain that this is precisely the worst-case.

Adjusting the budget using a constant $K$ may seem like a good idea, in principle, but take for example a control example: the pipeline here is composed of sensor, control, and actuator. If we increase the budget of the control task with $K=2$, this means that we are trying to consider two different sensor messages in the same controller execution. However, the result is that there is only one actuation command that can be applied at the end of the budget, and hence the two tasks would overlap. More in general, once the budget of one task in the pipeline is inflated, this also has consequences on all the tasks that are connected to it in the pipeline. I don't see therefore how it is possible to calculate a "per task" $M_i$ as in Equation (9). Is the new loss rate calculation taking into account the fact that tasks overwrite the same values and maybe the values are then not meaningful any longer?

As a minor note, it is quite annoying that the same concept changes name (in Section 4.1 there is $K$ and then right after in Equation (9), $M_i$ appears).

The paragraph that starts with "Our heuristic constraint solver algorithm [...]" is very unclear. I don't really understand what this means, with two forward references and very little information.

The evaluation is conducted with independent tasks while pipeline examples have been made available, for example in "Real world automotive benchmarks for free" (Simon Kramer, Dirk Ziegenbein, and Arne Hamann) or also the RTSS 2021 challenge (http://2021.rtss.org/wp-content/uploads/2021/06/RTSS2021-Industry-Challenge-v2.pdf). Overall the evaluation seems to be superficial and rather a comparison of different solvers than something that provides insights into the problem and solution.


**Expertise:**

Actively publishing in this area

**Useful:**

no

---

### Official Review · Reviewer_WrPC · 2021-11-27
**The paper has merit but is not free from some limitations as listed below— if addressed, the paper could be considered for publication.**

**Decision:**

Weak reject: interesting papers with flaws, not sure if they can be fixed in three months

**Review:**

This paper presents delay and loss-rate aware scheduling of tasks in cause-effect pipeline chains. The paper is pretty well-written, easy to follow, and solves a practical problem. The analytical formulations also seem rigorous. However, there are two fundamental limitations of the paper.

First, the uniprocessor case: This is the classic design-time schedulability analysis. While the analytical part is convincing, the evaluation results show CoPi performs very poorly, especially the lost rate bound <= 25%. The authors need to justify the requirement of such a heuristic approach, given that state-of-the-art solvers can find a solution with a high acceptance rate. Since this is a purely design-time test, the runtime overhead (e.g., Table 2) should not be an issue in this case.

Second, the multiprocessor case: the way the paper is written, first, it seems the heuristic is about design-time analysis used in traditional real-time schedulability tests. However, the authors then introduce the runtime pipeline optimization method (RPO) in Section 6.2, making the model less plausible and leaving many issues undiscussed. For instance, (a) How often the Algorithm 2 will be executed and conduct RPO? (b) How will PRO be integrated into the real-time schedulers? (c) What are the criteria that trigger the new budget and periods?

Further, the runtime for a pipeline of 10 tasks is as high as 60 ms in a general-purpose Core i5 laptop. What type of real-time schedulers can tolerate 60 ms of delays at runtime? How will this significant overhead be feasible in embedded devices while retaining real-time guarantees if the scheduler needs to execute Algorithm 2 quite often to conduct RPO? In contrast, if RPO is not needed, traditional optimization solvers' timing overhead is not a problem. Then, why should the designers use CoPi given that traditional solvers can result in better acceptance?

Other issues:

The paper is evaluated on simulations/synthetic workloads only. The paper will be further strengthened if the authors include actual workload, case study, and an implementation/evaluation of Algorithm 2 in existing real-time schedulers. The simulation parameters are used without any justification. For instance, why N=10? Why LGB in [13, 15]?

There are typos. For instance, "Latecy" in the x-axis of Fig 8a.

Final comments: The scheduling problem addressed in this work is worth considering, and the paper has intellectual merit. The paper could be improved in the following ways:

(a) For the uniprocessor case, the authors need to justify why such a heuristic approach is reasonable compared to existing MINLP solvers. For instance, if the problem is NP-hard, the traditional solvers may break in certain conditions — the authors could analytically or experimentally show the efficacy of CoPi for such cases.

(b) For the multiprocessor case, address the issues mentioned above, and include the realistic implementation on existing schedulers. For instance, in the Introduction, the authors mentioned that ROS uses such a pipeline-based model — how could the designers integrate algorithm 2 RPO techniques in a ROS chain? (c) The evaluation part should be improved.  For example, the authors should also include a thorough evaluation with realistic workload/case-study/implementation on real hardware/real-time schedulers.  Further, justifications of the parameters used in the simulations/experiments should be presented precisely.

**Expertise:**

Actively publishing in this area

**Useful:**

yes

---

### Official Review · Reviewer_mVx9 · 2021-12-16
**heuristic scheduling optimization for pipeline tasks**

**Decision:**

Weak accept: good paper with flaws that can be fixed in three months

**Review:**

This paper deals with the scheduling problem for pipelined tasks, where there’s an overall end-to-end delay. For each task in the (linear) execution order, new computation times and periods are derived for data-independent periodic subtasks that can be scheduled under traditional RM, so as to guarantee end-to-end delays and acceptable loss rates. In simulation, the heuristic approach, called CoPi, outperforms MINLP solvers.

In section 3.2 the authors write “Based on the relationships between the periods of all the producer-consumer pairs starting from the source to the sink task, the end-to-end loss rate of a pipeline is calculated.” This statement is pretty vague… the authors should explain how this computation is performed.

Soon after equations 6,7,8 it’s stated that the problem is an integer nonlinear programming problem. This is a bit unclear to me just from the equations, where max could be handled with integer constraints, but I do not see nonlinear arithmetic involved. Where specifically do the nonlinear terms come from. Is the difficulty of this problem that the periods and computation times are assumed to be integers? I suspect scheduler granularity is fine enough that real-valued solutions would be acceptable. This could be evaluated.

In Section 5.1 the authors mention in [23] a similar method is used. Further clarity could be provided on the differences between these two, or a comparison could be done in the evaluation with the method in [23].

In section 6.1 dealing with migration overhead and table 5, it doesn’t seem like the cost of migrations is reflected anywhere in the scheduling algorithm, only that task migrations can be kept low with the proposed approach.

In the evaluation in section 7, the approach is evaluated on 1000 randomly generated task pipelines. How are these created and are they realistic? Evaluating realistic pipeline parameters would provide a better evaluation than random parameters, which can be misleading for an evaluation.


**Expertise:**

Follow the literature closely, last published 5+ years ago

**Useful:**

yes

---

### Official Review · Reviewer_iwZk · 2021-12-19
**The paper does not demonstrate the use of CoPi for any real-world applications.**

**Decision:**

Weak accept: good paper with flaws that can be fixed in three months

**Review:**

Paper summary: Embedded applications consists of one or more sensor-to-actuation pipelines, e.g., object perception pipelines in autonomous vehicle systems. These pipelines are typically modelled as multiple data-dependent tasks (or processes). They have strict end-to-end (E2E) constraints because of the cyber-physical nature of such applications. Additionally, the pipelines may also have constraints on the end-to-end sampling rates, which determine the quality of service provided. When deploying such pipelines on multiprocessor embedded platforms, mapping the tasks to appropriate scheduling policy, parameters, cores such that the constraints are satisfied is known to be an NP-hard problem, even if the the workload is known in advance (as is the case in many embedded applications). While prior works have proposed to use constraint solvers to come up with a satisfiable task schedule, these are slow, especially when the workload changes at runtime and new schedules are desired at runtime, This paper proposes a heuristic-based algorithm instead — CoPi — that maps the pipeline to independent tasks and data buffers, and uses rate-monotonic scheduling to come up with a set of feasible schedule. CoPi is evaluated against open-source constraint solvers like GEKKO, pyomo, and scipy using pipelines with synthetic parameters. CoPi outperforms promo and scipy. While it is slightly worse than GEKKO in terms of scheduling capabilities, it finds a solution much faster than GEKKO.

Dear authors, thank you for submitting your paper to JSys! Thank you also for anonymously sharing the CoPi source!

I enjoyed reading about CoPi. The paper is written well; the CoPI algorithm is particularly nicely explained. I believe CoPi can be very useful in many modern-day embedded applications and it also has the potential to be used for applications outside the embedded systems domain, e.g., datacenter systems with QoS constraints. The paper also presents an extensive evaluation for CoPi against open-source baselines for a large parameter space. My main concern is that the paper does not demonstrate the use of CoPi for any real-world applications, i.e., there are no case studies using realistic workloads. In the absence of such case studies, I expect that at least the constraints are directly derived from real-world applications. While this may be the case, this is not apparent from the paper. For example, the paper currently does not sufficiently show if runtime scheduling is a problem in applications and if yes how does a certain scheduling delay fit within the overall scheme of things. In the following, I summarize the strengths and weaknesses of the paper in brief, followed by detailed comments.

Strengths:

+ The problem is relevant for many modern embedded applications
+ Given the QoS constraints in many datacenter applications today, the paper also highlights possible design choices for scheduling in datacenter systems
+ The solution is explained clearly with underlying formalism, wherever needed
+ The solution is evaluated extensively against multiple open-source constraint solvers and shown to outperform them in terms of scheduling capabilities and/or speed
+ Overall, the solution seems to be simple and intuitive, and easily deployable in practical systems
+ CoPi is open source!

Weaknesses:

- CoPi is motivated from real-world pipelines, but the paper does not demonstrate the use of CoPi for a real-world application
- While runtime scheduling is a motivation, the paper does not delve deeper into such examples from real-world applications, neither does it show what types of constraints exist in practical systems
- There seems to be an unnecessary emphasis on loss rates in the early part of the paper
- The paper assumes the reader is familiar with many concepts from real-time systems literature

Detailed comments:

* The phrase “loss-rate” is used multiple times in the Introduction and Abstract. However, despite being familiar with the real-time systems literature, I had a hard time understanding the exact meaning of this phrase. I suggest the meaning is clarified once in the beginning itself.

* The paper never really explains why the design avoids timing and data dependencies, and what does asynchronous scheduling mean (as opposed to synchronous scheduling before). I personally understand the intuition and the rationale, but it may not be obvious to all readers, especially when reading the following sentences in the introduction. “The main idea behind CoPi is to get rid of the unnecessary delay and message losses in a pipeline, and consequently avoiding the timing and data dependencies between the communicating tasks [21].”, “As CoPi meets the E2E delay and loss-rate guarantees of a pipeline, the tasks are asynchronously scheduled without any timing or data dependencies between each other.”

* Explain the WFD heuristic and the Simpson's four-slot algorithm in brief, maybe in footnotes.

* “We show that CoPi performs significantly better, an order of magnitude at the highest, in runtime, and comparably in acceptance ratio, with respect to other solvers.” Simplify this statement. What is acceptance ratio?

* “Without the loss of generality, we consider unidirectional pipelines.” Please elaborate. What about cycles?

* Could you cite examples where L (loss rate) is an input parameter of the system?

* “If a task has already finished its work for a job invocation, it yields and does not start its next job until next period.” When and why does this happen?

* “Moreover, the fixed execution strategy tightly bounds a pipeline’s end-to-end latency [26].” What does a fixed execution strategy mean?

* I realize the Equation 1 is derived from prior work. I would still prefer to read about how it is derived in this paper. “As Equation 1 is a recursive equation, the computation time depends on the wanted precision on E.” Why is the equation recursive? Is it because Ri’s are unknown? The LHS E does not feature on the RHS.

* “Aligned with Feiertag et al.’s concept of data-path reachability conditions [21], the ratio of non-reachable messages to the total number of messages is the loss-rate of a pipeline.” I found this reference unnecessarily confusing. Reachability conditions can be interpreted as something completely different.

* At the beginning of Section 3.2, it was unclear whether loss rate refers to only messages that are input to the source task, or to all tasks.

* Does the ratio in Equation 4 also hold if the two tasks are not started synchronously? Does this equation represent an upper or lower bound on the sampling ratio?

* “Then, the loss-rate of a producer-consumer pair is …” I think the example in this sentence is wrong.

* I think Section 3.2.1 can be significantly shortened. Particularly, the proofs in the section are really just explanations. I suggest keeping the rules as they are, but getting rid of the proofs, and weaving the explanations along with the examples, which are most helpful in understanding the concepts.

* “This problem is known to be NP-hard [29].” Citations [29] is a general paper on nonlinear integer programming. Could you also cite a paper from the real-time systems literature. Recently, many papers have proposed precise complexity classes for different problems in the RTS domain. Could you cite one of these papers, if applicable?

* The rationale for budget adjustment was not clear in Section 4.1. It became clear in Section 5 later.

* In equations 9 and 10, M is used as multiplier, but in the paragraph earlier, K is used.

* “Gerber et al. also proposed a similar approach of deriving task periods, offsets and other parameters from the end-to-end constraints, albeit on task precedence relations [23].” How are task precedence relations different from the dependencies considered in the paper?

* Runtime migrations may affect the budget parameter C. The paper currently does not comment on this aspect.

* “constraints constraints” —> “constraints”

* “CoPi takes the initial task runtime budgets (C in the model and budgets in Algorithm 1) and the desired upper bound on the end-to-end delay (EUB in the model and e2e_ub in Algorithm 1) as its inputs.” What about the upper bound on the loss rate?

* “CoPi runs Stage 1 only once for a pipeline, but it runs Stage 2 and 3 multiple times with different $\alpha$ values.” Can we instead start with one large $\alpha$?

* Figure 9(c): it would be nice to also learn about how GEKKO’s runtime varies with the pipeline length.

**Expertise:**

Published in this area in the last 5 years

**Useful:**

yes

---

### Official Review · Reviewer_aocA · 2021-12-23
**The paper is a contribution to scheduling Real-time task pipelines as a set of asynchronous, periodic tasks on a processor as well as extending it to schedule multiple task pipelines on a multiprocessor system. This fills a gap in the scheduling literature of pipeline-based task models. The paper presents an analytical framework with design-space exploration that can be intergrated into a middleware such as ROS or a Multi-core OS.**

**Decision:**

Strong accept: excellent paper that will help the community

**Review:**

Summary : This paper presents an algorithm to schedule Real-time periodic tasks that are expressed as pipelines. The Real-time requirements here include end-to-end (E2E) delay and loss-rate guarantee of the pipeline. The algorithm returns the budgets and periods that each of the tasks get during runtime at every stage of the pipeline. The resulting tasks with the budgets and period can the be scheduled asynchronously and without any data dependencies among them.

The paper also has takes another step forward and maps tasks to a Multiprocessor platform. The tasks are mapped to processors using a heuristic. For dynamic pipelines, it also does - (a) runtime task migration  and (b)scheduling parameter optimization

Pros
- The system model is well-presented,
- Visualizations of the algorithm operation provided improve the readability of the paper.
- Reduction of utilization for already accepted task pipelines when mapping to multi-processor fails is a tradeoff between end-to-end delay and utilization
- Comparison with state of the art solvers that support integer non-linear programming is well-presented
- The paper has released its artifacts with a good README file, kudos to the authors
- Pipelines are common in sensing-actuation chains as seen in automotive and avionics systems, and applications such as ROS can use CoPi to predictably schedule pipelines using this scheme

Cons
- The schedulability used in RMS for single processor and partitioned RMS for multiple processor, so the utilization is still bounded by the theoretical limits of RMS.
- The disadvantage of scaling the period and adjusting the computation time can lead to increased memory requirements of buffers between tasks, the overhead of which is not presented
- the values of $\alpha$ and $\beta$ are empirically determined with no application mentioned for reference
- Task assignment and migration on multiprocessor can be made practical by considering the system implementation details, but these have elided

Questions
- Why is LBG (Latency Bandwidth Gap) plotted only for values between 10 and 15? It would be useful if the authors can discuss this.
- Why does the loss rate = 25% lead to loss of performance of CoPi (Fig 8b)? This can be discussed in camera-ready
Other comments
- The authors mention about integrating the framework into a Multi-core OS and into the ROS middleware, these will greatly enhance the impact of this work.


**Expertise:**

Published in this area in the last 5 years

**Useful:**

yes

---

### Meta-Review · Area_Chair_Q3ZD · 2022-01-06

**Recommendation:** Revise
**Confidence:** 5

**Metareview:**


The main concerns for the reviewers are listed here. If the authors can address all of these issues (and the details in the full reviews) in the given timeframe, then the paper has a high chance at acceptance:

1. Lack of evaluation with real-world applications or a discussion of whether the evaluation parameters used in the paper are derived from actual requirements. This is particularly problematic since there exist pipeline examples from the community, viz:

    (a) "Real world automotive benchmarks for free" (Simon Kramer, Dirk Ziegenbein, and Arne Hamann

    (b) RTSS 2021 challenge (http://2021.rtss.org/wp-content/uploads/2021/06/RTSS2021-Industry-Challenge-v2.pdf)

2. The theoretical models need refinement/better explanations:

    (a) For the uniprocessor case, the authors need to justify why such a heuristic approach is reasonable compared to existing MINLP solvers

    (b) For the multiprocessor case, including the realistic implementation on existing schedulers need to be explained

    (c)  using the correct bounds (e.g., without offsets) will simplify a lot the equations and potentially reduce the contributions of the paper — so the authors should consider this carefully and explain their contributions better once they have fixed the problems with the equations

3. Missing comparisons with related work, e.g. papers like "Time-selective data fusion forin-network processing in ad hocwireless sensor networks" (by Jaanus Kaugerand, Johannes Ehala, Leo Motus, and Jurgo-Soren Preden)

Other issues:
1. practicality issues: the runtime for a pipeline of 10 tasks is as high as 60 ms in a general-purpose Core i5 laptop. For embedded systems this could be higher, thus seriously affecting the performance.
2. How is the nonlinear arithmetic involved? Where specifically do the nonlinear terms come from

Additional suggestions on how to improve the paper:
1. equations (1) and (2) can be fixed in writing -- particularly clarifying references [20], [26] and [48]. The authors need to clarify that they have used that equation from [26] which does not include offsets. They can also add a numeric example to make equations (1) and (2) more readable. For example, they have done well in Section 3.2.1 to explain sampling ratio equation (for the oversampling and undersampling case)
2. the definitions of loss rate need to be explained better viz., clarifying the distinction between message and task. messages have not been defined until this point. Perhaps draw on the definition from [21].
3. the authors could add a discussion section before Related Work and after Evaluation (call it Sec. 7) to recommend how their scheme CoPi can be used in an application such as ROS.
4. The x-axis in LBG (Fig 8) is only varied from 11 to 18. Is this drawn from practical examples? If so, this needs to be clarified. Also, the process of deducing usable values of α and β
5. Fix consistency of notation between K and M issues in Section 4.1 on Budget assignment, which can easily be fixed in writing.
6. Fix grammar and writing issues as pointed out by a couple of reviewers.

---

### Decision · Program_Chairs · 2022-01-06

**Decision:**

Accept

**Comment:**

Dear authors,

Based on the reviewers and a subsequent discussion, we have settled on a revised-and-resubmit decision for your paper. The reviews and meta-review will be published later today. As per JSys policy, you have up to three months to submit your revised paper. Please highlight the changes in the revision to facilitate the second round of reviewing.

Once again, we apologize for the delay in communicating our decision.

Update: the paper has now been accepted